



# Pollen-based temperature and precipitation changes in the Ohrid Basin (western Balkans) between 160 and 70 ka

Gaia Sinopoli[1,2,3], Odile Peyron[3], Alessia Masi[1], Jens Holtvoeth[4], Alexander Francke[5,6], Bernd Wagner[5], Laura Sadori[1]

[1]Dipartimento di Biologia Ambientale, Sapienza University of Rome, Rome, Italy.
[2]Dipartimento di Scienze della Terra, Sapienza University of Rome, Rome, Italy.
[3]Institut des Sciences de l'Evolution de Montpellier, University of Montpellier, CNRS, IRD, EPHE, Montpellier, France.
[4]Organic Geochemistry Unit, School of Chemistry, University of Bristol, Bristol, UK.
[5]Institute of Geology and Mineralogy, University of Cologne, Cologne, Germany.
[6]Wollongong Isotope Geochronology Laboratory, School of Earth and Environmental Sciences, University of Wollongong, Wollongong, Australia.

*Correspondence to*: Alessia Masi (alessia.masi@uniroma1.it)

**Abstract.** Our study aims to reconstruct climate changes that occurred at Lake Ohrid (south-western Balkan Peninsula), the oldest extant lake in Europe, between 160 and 70 ka (covering part of Marine Isotope Stage – "MIS" - 6 and all of MIS 5). A multi-method approach, including the "Modern Analogues Technique" and the "Weighted Averaging Partial Least-Squares Regression", is applied to the high-resolution pollen sequence of the DEEP site, collected from the central part of Lake Ohrid, to
provide quantitative estimates of climate and bioclimate parameters. This allows us to document climatic change during the key periods of MIS 6 and MIS 5 in South Europe, a region where accurate climate reconstructions are still lacking for this time interval.
Our results for the penultimate glacial show cold and dry conditions, while the onset of the Last Interglacial is characterized by wet and warm conditions, with temperatures higher than today (by ca.
2°C). The Eemian in the Balkans was not a stable phase and a climatic tri-partition, with an initial phase of abrupt warming (128-121 ka), a central phase with decreasing temperatures associated to wet conditions (121-118 ka), followed by a phase of progressive change towards cold and dry conditions (118-112 ka), is evident.
After the Eemian, an alternation of four warm/wet periods with cold/dry ones, likely related to the
succession of Greenland stadials and cold events known from the North Atlantic, occurred. The observed pattern is also consistent with hydrological and isotopic data from the central Mediterranean. The Lake Ohrid climate reconstruction shows greater similarity with climate patterns inferred from northern European pollen records than with southern European ones, which is probably due to its intermediate position and the mountainous setting. However, this hypothesis needs further testing as
very few climate reconstructions are available for southern Europe for this key time period.





## 1 Introduction

Since the Mid Pleistocene, the Quaternary is characterized by high-amplitude glacial-interglacial climate variability, occurring cyclically with a 100 ka (kiloanni) periodicity (e.g. Raymo et al., 1989; Tzedakis et al., 1997). The marine isotope stages - "MIS" - 6 (glacial) and 5 (interglacial-glacial) are

defined by marine oxygen isotope records ($\delta^{18}O$; Lisiecki and Raymo, 2005). MIS 6 is also named the penultimate glacial (or Riss Glaciation) and can be roughly dated from ca. 190 to 130 ka, while MIS 5 is termed the Last Interglacial Complex (LIC) and lasts from ca. 130 to 80 ka (Govin et al., 2015; Railsback et al., 2015 and references therein). The penultimate glacial is characterized by a millennial-scale climate variability (Martrat et al., 2004), and the end of MIS 6 is characterized by several abrupt

events, which are probably related to the iceberg-rafted debris (IRD) deposition intervals in the Northeast Atlantic (McManus et al., 1999). In contrast, the LIC includes the Last Interglacial (roughly equivalent to MIS 5e, or Eemian), followed by a period named "Early Last Glacial" characterized by a succession of stadial (cold/dry conditions) and interstadial (warm/wet conditions) periods (MIS 5d to 5a). These stadials and interstadials correlate to glacial advances/retreats that are documented by ice-

rafted debris in North Atlantic sediments (e.g. Bond events and Heinrich events: Bond et al., 1992; Bond and Lotti, 1995) and by changes in oxygen isotope composition in Greenland ice cores (Dansgaard-Oeschger cycles: Dansgaard et al., 1993). Equivalent to the marine isotope stages, the ice core records distinguish Greenland stadials (GS) and Greenland interstadials (GI) where short-lived cold episodes are associated to surface ocean cooling (C events). Across the LIC, seven such cold

events (C19-25) have been documented (McManus et al., 1994; Oppo, 2006; Rasmussen et al., 2014). The Eemian (127-110 ka, Turner 2002, Shackleton et al., 2003) is of particular interest with regard to orbital parameters inducing a strong seasonal forcing of insolation, contrasted vegetation changes (Beaulieu and Reille, 1992; Zagwijn, 1996) and climatic conditions (Cheddadi et al., 1998; Sánchez-Goñi et al., 2012). Therefore, this period is also considered as a useful target for General Circulation

Models (GCMs) data-model comparison (Kaspar et al., 2005; Otto-Bliesner et al., 2013). In the Northern Hemisphere, the Eemian was wetter (Fauquette et al., 1999; Guiot, 1990; Guiot et al., 1993; Klotz et al., 2003) and warmer in summer by up to 1-2 ºC than the Holocene (Kaspar et al., 2005; Otto-Bliesner et al., 2013; Overpeck et al., 2006;), while sea level was ca. 6 - 9 meters higher (e.g., Kopp et al., 2009). The Eemian thus allows to study climate dynamics and ecosystem response in a warmer than

present Northern Hemisphere without the influence of anthropogenic activity, thereby contributing to assessments of the future impact of the current anthropogenic climate change. Earlier studies of the Eemian considered it a stable, uninterrupted warm period (e.g. Guiot et al., 1992; McManus et al., 1994; Pons et al., 1992; Zagwijn, 1996), with climatic oscillations only recorded in the final part, at the transition with the following glacial, i.e. the Early Würm (Field et al., 1994; Litt et al., 1996). However,

more recent studies suggest that low amplitude climatic fluctuations did occur during the Eemian (e.g. Brewer et al., 2008; Sanchéz-Goñi et al., 2005; Sirocko et al., 2005) and in North GRIP ice core isotope records (NGRIP Members, 2004). A pronounced short-lived climatic fluctuation, the intra-Eemian cold event, occurred around 122 ka (Maslin and Tzedakis, 1996). Climate changes across the penultimate glacial and the Eemian are documented by numerous pollen records from marine and terrestrial archives

(e.g. Govin et al., 2015; Kaspar et al., 2005; Otto-Bliesner et al., 2013). Some of these records have been used for the reconstruction of climatic parameters with a quantitative approach synthesized in



Brewer et al. (2008). However, most of these have been carried out using pollen data from European sites located north of 45°N, while only few reconstructions were carried out in southern Europe. Two are based on pollen continental records, Lago Grande di Monticchio in southern Italy (Allen et al., 2000) and Ioannina in north-western Greece (Tzedakis, 1994) and two on marine pollen records, MD
99-2331 and MD 99-2042 on the Iberian margin (Sánchez-Goñi et al., 2005, Brewer et al., 2008). A first North-South comparison suggests that the two regions may have experienced a somewhat different climatic pattern during the Eemian (Brewer et al., 2008). While both regions experienced an early temperature optimum followed by a cooling trend, towards the end of the Eemian, temperatures and precipitation decreased more strongly in northern Europe compared to southern Europe (Brewer et al.,
2008; Sánchez-Goñi, 2007 and references therein). Given that this comparison is based on 13 North European sites and on only 4 South European sites there is a need to provide more reliable quantitative climate reconstructions in southern Europe for the Eemian and for the entire Last Interglacial Complex in order to improve our understanding of the climate response during high-amplitude glacial-interglacial cycles.
The Balkan Peninsula is unambiguously a key region at the confluence of central European and Mediterranean climate influences. The area is rich in extant Quaternary lakes and palaeolakes, with sediment records providing essential information on past vegetation and climate changes going back hundreds of thousands years, such as Lake Ohrid (Albania/F.Y.R.O.M.: e.g. Lézine et al., 2010; Sinopoli et al., 2018; Wagner et al., 2017), Lake Prespa (Albania/F.Y.R.O.M./Greece: Panagiotopoulos
et al., 2014), Ioannina (West Greece: Tzedakis et al., 2003), Tenaghi Philippon (North-East Greece: Milner et al., 2016), and Kopais (South-East Greece: Tzedakis, 1999; Okuda et al., 2001). Despite the richness in long palaeoenvironmental archives, quantitative palaeoclimatic reconstructions have been rarely attempted or cover relatively short periods.
In this study, we use a multi-method approach to reconstruct climate parameters between the end of the
penultimate glacial (160 to 128 ka) and the Last Interglacial Complex (128-70 ka) inferred from the exceptionally long palynological record (569 m) of the Ohrid Basin in the Western Balkans (Sadori et al., 2016). The high-resolution palynological data has been acquired from a sediment core from the center of the lake (DEEP site: Sinopoli et al., 2018). The approach includes two methods frequently used in palaeoclimate reconstructions: the Modern Analogs Technique (MAT: Guiot, 1990) and the
Weighted Averaging Partial Least-Squares Regression (WAPLS: Ter Braak and Juggins, 1993). In order to test the reliability of our numerical approach, we compare the results to independent climate proxies from the Ohrid Basin such as biomarkers (Holtvoeth et al., 2017) and total inorganic carbon (TIC) concentrations, which largely represent authigenic calcite precipitation (Vogel et al., 2010; Francke et al., 2016). To discuss the climate signal at a more global scale, we compare our results to
available pollen-based reconstructions from northern Europe and the Mediterranean, and to marine and terrestrial proxies from the Mediterranean and the Northern Hemisphere (e.g., De Abreu et al., 2003; Drysdale et al., 2005; NGRIP Members, 2004; Lisiecki and Raymo, 2005; Martrat et al., 2014; Regattieri et al. 2014, 2017; Sánchez-Goñi et al., 1999; Wang et al., 2010).



## 2 Site description

A complete list of supporting literature can be found in Sadori et al., 2016, Sinopoli et al., 2018, and Wagner et al., 2017. Lake Ohrid is located on the Balkan Peninsula at the border between the Former Yugoslav Republic of Macedonia (F.Y.R.O.M.) and Albania (Fig. 1). It is probably the oldest lake of

Europe, with an estimated age of 1.2 Ma. The lake has a tectonic origin as its catchment is located in a graben that formed during the Alpine orogenesis between ca. 10-2 Ma ago. Today, Lake Ohrid has a surface area of 360 km$^2$ (30 km long, 15 km wide, 693 m a.s.l.), an average depth of 164 m and a maximum depth of 293 m. The basin is bordered in the West by the Mokra Mountains (1514 m a.s.l.) and in the East by the Galičica Mountains (2265 m a.s.l.). The latter separate the watersheds of Lake

Ohrid and adjacent Lake Prespa (849 m a.s.l.), which is located ca. 10 km to the East, although the two lakes are connected via a karst aquifer system. Apart of inflow from Lake Prespa via the karst aquifers, Lake Ohrid is supplied with water from surface run-off via small streams and rivers and by direct precipitation. Modern climate in the Ohrid region is Mediterranean with continental influences. The thermal capacity of the lake as well as its proximity to the Adriatic Sea and the local topography affect

the local climate. The mean annual temperature recorded in the Ohrid region averages at 11.5 °C; temperatures range between ca. 2 and 6 °C in winter (minimum in January) and between 10 and 22 °C in summer (maximum in July). The morphology of the catchment also affects the wind regime, with northerly winds prevailing during winter and southerly and southeasterly winds during spring and summer. The pluviometric regime is Mediterranean, with an average annual precipitation of 878 mm

(Fig.1).
Lake Ohrid has a rich macrophytic flora (more than 124 species) distributed into four zones dominated by *Lemna trisulca* L., *Phragmites australis* (Cav.) Trin. ex Steud., *Potamogeton* L., Characeae, *Ceratophyllum* L., *Myriophyllum* L. and the colonial alga *Cladophora* spp. The present vegetation around Lake Ohrid belongs to the sub-mediterranean type, in which Mediterranean and Balkan elements

dominate together with central European ones. The vegetation is sequenced in altitudinal belts, starting from lake level (693 m a.s.l.) to the top of the mountains (ca. 2200 m a.s.l.). Riparian forest (dominated by *Salix alba* L.), with elements of mediterranean vegetation (*Fraxinus ornus* L., *Pistacia terebinthus* L. and *Phyllirea latifolia* L.), is present from the altitude of the lake level to lower elevations together with *Buxus sempervirens* L., *Quercus trojana* Webb, *Carpinus orientalis* L. and *Ostrya carpinifolia* Scop.

Otherwise, forests are characterized by mixed deciduous elements and are mainly composed of *Quercus cerris* L., *Q. frainetto* Ten., *Q. petraea* (Matt.) Liebl., *Q. pubescens* Willd. up to 1600 m a.s.l., followed by montane and mesophilous forests (from 1600 to 1800 m a.s.l.) dominated by *Fagus sylvatica* L. in association with *Carpinus betulus* L., *Corylus colurna* L. and *Acer obtusatum* (*Acer opalus subsp. obtusatum* (Waldst. & Kit. ex Willd.) Gams). *Abies alba* Mill. and *A. borisii-regis* Matt. mixed forests

grow below 1900 m a.s.l., at the upper limit of the forested area, while sub-alpine grassland and shrubland with *Juniperus excelsa* (subsp. *polycarpos* (K. Koch) Takhtajan) are found above the tree-line in mountains situated at south-east of the lake. Towards the East, *Pinus peuce* Griseb., is present at high elevation, associated with *Pteridium aquilinum* (L.) Kuhnor, *Vaccinium myrtillus* L. Sparse population of *Pinus* species considered to be Tertiary relics are present in the wider area.


Suspendisse a elit ut leo pharetra cursus sed quis diam. Nullam dapibus, ante vitae congue egestas, sem ex semper orci, vel sodales sapien nibh sed lectus. Etiam vehicula lectus quis orci ultricies dapibus. In sit amet lorem egestas, pretium sem sed, tempus lorem.

## 3 Materials and Methods

### 3.1 Pollen data from the DEEP core

A drilling campaign mainly funded by the International Continental Scientific Drilling Program (ICDP) was carried out as part of the project Scientific Collaboration On Past Speciation Conditions in lake Ohrid (SCOPSCO) in 2013. Six parallel cores were recovered from the depocenter of the lake at 243 m water depth (DEEP site). A composite sequence representing an overall sediment depth of 569 spanning at least the last 1.2 million years has been obtained (Wagner et al., 2017). According to the age model, the uppermost 247.8 m of the DEEP core cover the last 637 ka (Francke et al., 2016). Palynological data has been published for the upper 200 m of the DEEP pollen record, covering the last 500 ka, with a time resolution of ca. 1600 years (Bertini et al., 2016; Sadori et al., 2016). Results have shown an alternation of forested and non-forested periods that are ascribed to five glacial-interglacial cycles. The study presented herein is based on new high-resolution pollen data (one sample every ca. 400 years, Sinopoli et al., 2018) summarised in Figure 2.

### 3.2 Quantitative reconstruction of temperature and precipitation

We adopted two different methods in order to improve the error assessment (e.g., Klotz et al., 2003; Kühl et al., 2010; Peyron et al., 2005, 2011, 2013). It has been demonstrated by several studies that reconstructions based on just one method can have limitations, depending on the time interval and on the methods chosen (Birks et al., 2010; Brewer et al., 2008). Here, we have selected the Best Analogs Approach or Modern Analog Technique (MAT: Guiot, 1990) and the Weighted Average Partial Least-Squares Regression (WAPLS: Ter Braak and Juggins, 1993), two classical methods already used to reconstruct climate changes in the Mediterranean during the Holocene and other time periods (e.g. Brewer et al., 2008, Mauri et al., 2015; Peyron et al., 2011, 2013). Both methods are based on the assumption that climate change strongly influences the distribution and composition of vegetation as every plant species tolerates distinct ranges of temperature and humidity. The MAT is based on the comparison between fossil pollen assemblages and modern ones. The MAT determines the degree of dissimilarity (in terms of taxa abundance and composition) between modern pollen data (associated to known climatic parameters) and the fossil data for which the climatic parameters are to be estimated. For each fossil pollen assemblage, a number of modern pollen assemblages are selected (based on a chord distance calculation) as the closest ones or "analogues". In the present paper, we have chosen a number of six modern analogues. The method uses the present-day climate data associated to the selected modern analogues to infer the past climate values (Guiot, 1990). In contrast to the MAT, the WAPLS method is a transfer function, which uses a real statistical calibration between climate parameters and modern pollen data. The method is based on unimodal relationships between pollen percentages and climate. In WAPLS, several components are calculated based on weighted averaging





algorithms that successively explain more variance in the data; this means that taxa, which better define a climate parameter are weighted more than the other ones (Ter Braak and van Dam, 1989). A cross-validation has been done for determining the right number of components (Ter Braak and Juggins, 1993). For both methods, we have used a modern pollen dataset containing more than 3088 samples
from European and Mediterranean regions (Peyron et al., 2013). From this dataset, we have excluded those pollen samples collected in warm to hot steppes in order to improve the climate reconstruction during steppic phases (Tarasov et al., 1998). Moreover, *Pinus* has been excluded due to its overwhelming presence in the DEEP record that potentially masks climatically controlled environmental signals from other taxa.
Five climate parameters have been reconstructed for the DEEP pollen record (Fig. 3) with each method: 1) the mean temperature of the coldest month (MTCO), 2) the mean temperature of the warmest month (MTWA), 3) the mean annual temperature (TANN), 4) the mean annual precipitation (PANN) and 5) the growing degree days above 5°C (GDD5) (Peyron et al., 1998). The analysis was carried out with the software package R program, a system for statistical computation and graphics (R Foundation,
https://www.r-project.org/), by using the Package 'rioja' (Juggins, 2016). Error bars have been calculated but are not shown in the figure for graphic clarity. They are available in the Appendix (Fig. A1). Fig. A2 (Appendix) indicate the reliability of the analogues selected by reporting the squared-chord distance between the first and the last analogue for a chosen climate parameter (TANN) calculated by MAT method.

**4 Results**

Previous data show that MIS 6 was characterized by prevalence of *Artemisia*, Amaranthaceae and Asteroideae since 160 ka (Sadori et al., 2016). During the LIC, high resolution data provides evidence for forested periods (interglacial and interstadials) alternating with periods of a more open environment (stadials). The pollen analysis revealed that the surroundings of Lake Ohrid during the Eemian were
characterized by mesophilous communities prevailing on montane ones (Fig. 2). Forests were mainly featured by expansion of *Quercus robur* type and *Q. cerris* type together with *Pinus* and *Abies* (Sinopoli et al., 2018). Trees never completely disappear, being also recorded during stadial periods, albeit at low percentages. Here we adopt the terminology used by Woillard (1978) for La Grande Pile pollen record to enhance comparability (see Tab. 1). We are aware that the marine stratigraphy does not always
precisely match the terrestrial one (e.g. Sánchez-Goñi et al., 2007).
Our climate reconstruction suggests cold and dry conditions during MIS 6 and, in MIS5, an alternation of warm and wet conditions during the Eemian and St. Germain I and II interstadials with cooler and dryer ones occurring during stadials (Fig. 3).

**4.1 The late part of the Penultimate glacial (MIS 6, 160-128 ka)**

Quisque cursus massa sed urna congue, ac convallis neque consectetur. Proin faucibus neque non metus mollis, suscipit The late and long-lasting part of MIS 6 was very cold and dry (Fig. 3) as suggested by the results from both MAT and WAPLS. This portion of the glacial period can be divided into a first part, between 160 and 143 ka, which is characterized by cold and dry climate conditions (mean annual

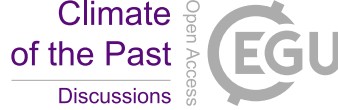

temperature below 6 °C and annual precipitation between 350 and 600 mm/yr), and a second part, lasting from 143 to 128 ka, when several short-term high-amplitude oscillations are reconstructed, especially from 140 to 135 ka. These abrupt changes involve all the climate parameters. These oscillations represent significant shifts in mean annual temperature (TANN) and precipitation (PANN),

ranging from 2.7 °C to 10°C and from 120 mm/yr to 600 mm/yr respectively. The GDD5 (growing degrees days over 5 °C, annual accumulated temperature over 5 °C) pattern is strongly linked to the MTWA (mean temperature of the warmest month) pattern. The pronounced peak in temperature around 138 ka is probably linked to the high percentages of mesophilous taxa; however, this increase seems overestimated with the MAT. Notably, this oscillation and also the other lower-amplitude oscillations

between 143 to 128 ka are more marked in the reconstruction inferred from the MAT (Fig. 3). These values are probably overestimated as the MAT is more sensitive than WAPLS and other methods (Brewer et al., 2008; Kühl et al., 2010). Brewer et al. (2008) demonstrated that a wider spread of estimates is found during colder periods and that the analogue methods seem to give a larger variability in time than the other methods, especially during the cold periods.

**4.2 The Last Interglacial Complex -LIG- (MIS 5, 128-70ka)**

**4.2.1 The last interglacial (128-112 ka) or Eemian**

At 128 ka, the transition to the last interglacial is marked by a rapid rise in temperature and in precipitation, being very close to modern values. The so-called thermal optimum of the Eemian, occurring at Lake Ohrid between 128 and 121 ka, is characterized by TANN between 10 and 12 °C, the

highest of the investigated period, 2 °C warmer than the present day (Figs 3, 5, 6). Winter temperatures were also warmer than today, while summer ones were close to modern values and precipitation 100 mm lower than present day (Figs 3, 5). A cool event is suggested between 121 and 118 ka, while precipitation reaches the highest values of the Eemian at around 119.4 ka. After this cool phase, during the last part of the last interglacial (118-112 ka), we reconstruct a progressive cooling and a decrease in

precipitation until the end of the Eemian at 112 ka.

**4.2.2 *The Early Last Glacial (112-70 ka)***

The Early Last Glacial (Table 1) is characterized by an alternation of short cold/dry periods with longer warm/ wet ones:
- warm and wet interstadial periods: St. Germain I (108-89.8 ka), St. Germain II (85.7-78.8), Ognon I

(76.6-75.4 ka), Ognon II (73.4-69.9 ka).
- cool and dry stadial periods: Melisey I (112-108 ka), Montaigu (105.2-104 ka), Melisey II (89.8-85.7 ka), stadial I (78.8-77.6 ka), stadial II (75.4-73.4 ka)

**4.2.2.1 Interstadials: warm and wet conditions**

The first interstadial following the Eemian (Fig. 3) corresponds to the St. Germain I (108-89.8ka) which

can be divided (Tab. 1) in three parts, two of which, St. Germain Ia (108-105.2 ka) and St. Germain Ic (104-89.8 ka), warm and wet. During St. Germain Ia, both TANN and PANN increase, suggesting that





the St. Germain I was warm and wet but less than the Eemian. In contrast, the St. Germain Ic appears to be wetter and overall warmer than St. Germain Ia (Fig. 3). The precipitation increases strongly and reaches values between ca. 600 and 900 mm/yr. It's the wettest period between 160 and 70 ka. A pronounced dry event is centered at 95.3 ka. The second interstadial (85.7-78.8 ka) corresponding to St.

Germain II is characterized by temperate conditions comparable to those of St. Germain Ic (104-89.8 ka) even if it seems drier. The following two interstadials corresponding to the Ognon I and II (77.6-75.4 and 73.4-69.9 ka) show climate conditions comparable to those occurring during the second interstadial (Fig. 3).

### 4.2.2.2 Stadials: cold and dry conditions

The temperate conditions of the last interglacial are interrupted by a first cooling event corresponding to the stadial Melisey I (112-108 ka) characterized by cold and dry conditions (Fig. 3). A second abrupt event is recorded between 105.2 and 104 ka, namely during the Montaigu cooling (St Germain Ib) that divides the St. Germain I into two "interstadials". During this event, precipitation reaches values similar to those of the previous stadial (Melisey I). Temperature and precipitation follow the same pattern with

a strong decrease at 104.6 ka. Melisey II (89.8-85.7 ka) appears as the coldest and driest event of the early last glacial (Fig. 3), with a strong temperature and precipitation decrease evidenced by both methods. As during the end of MIS 6, the cooling reconstructed with the MAT is probably overestimated given that the analogue method provides a large variability during the cold periods. However, the climate was certainly very cold during Melisey II, particularly in winter as illustrated by

the WAPLS. Precipitation reaches extremely low values, dropping to ca. 100 mm/yr, which is even lower than in MIS 6 but here too the drying seems overestimated with the MAT for the same reasons as temperatures. The following two stadials (78.8-77.6 ka and 75.4-73.4 ka) should indicate a pattern very similar to Melisey II, with an abrupt decrease of temperature and precipitation followed by a likewise abrupt increase at the end of each phase (Fig. 3).

## 5 Discussion

### 5.1 Differences between MAT and WAPLS and reliability of the methods

The temperature reconstructions from both methods are reasonably coherent (trends and values) during the interglacial and interstadials, but a wider spread of estimates is found during colder periods (Fig. 3) for which the analogues method suggests higher-amplitude oscillations and lower values than those

inferred by the transfer function (Fig. 3). More precisely, during the first part of MIS 6 (ca. 160-143 ka), both methods produce low-amplitude oscillations in temperatures, but the values determined by MAT appear to be around 4 °C lower than those determined by WAPLS. Even if the precipitation curves produced by both methods show the same trend, reconstructed values by MAT are roughly 300 mm lower than those resulting from WAPLS. After 143 ka, the differences between the two methods are

more marked (Fig. 3). It's worth to be mentioned that WAPLS precipitation values are inside the errors bars (Fig. A, supplementary materials). Discrepancies between the methods may be related to several factors that either depend on the method itself or on the composition of past pollen assemblages.





Modern analogues methods are very sensitive to minor variations in the pollen assemblages, especially during glacial periods (Brewer et al., 2008). Similar discrepancies associated with MAT also occur in the reconstruction of La Grande Pile; strong cold oscillations are evidenced (and probably overestimated) after the Eemian thermal optimum by Brewer et al (2008). MAT is frequently used to
reconstruct the climate of the Lateglacial and Holocene (e.g. Mauri et al., 2015; Peyron et al., 2005) but, as demonstrated by Guiot et al. (1993), ambiguous outcomes may occur particularly for past glacial and cold intervals (stadials). The major problem appears to be the lack of modern analogues or only limited similarity with past glacial vegetation (Guiot et al., 1993; Peyron et al., 1998). Indeed, as reported in several studies (Guiot, 1987; Guiot et al., 1993; Klotz et al., 2003), glacial steppe vegetation dominated
by high percentages of Amaranthaceae (as at Lake Ohrid, Fig. 2) has no present-day analogue in Europe. For this reason, we have used modern samples from cold steppe principally from the Tibetan Plateau and from Russia as "potential" analogues for glacial periods (Peyron et al., 1998, 2005). Squared-chord distance has been used to determine the degree of dissimilarity (Fig. A, supplementary materials), revealing that our reconstruction can be judged reliable and without a no-analogs situation
occurring. The differences between the two methods for cool/cold periods may also be ascribed to the quasi-continuous presence of arboreal taxa in steppic assemblages. During the period between 143 and 128 ka, the major oscillations are probably overestimated and likely linked to the presence of arboreal mesophilous (temperate) taxa in steppic pollen assemblages. Mesophilous taxa amount to 10-30 %, with prevalence of deciduous and semi-deciduous oaks, while pioneer shrubs are between 5-10 %, with
prevalence of *Juniperus* (Fig. 2). This discrepancy attests to the specific local hydroclimatic features of the Ohrid Basin and its fundamental role as a refugium for many arboreal taxa. Considering the very high sensitivity of the MAT, WAPLS seems to be a better method to reconstruct the climatic changes during cold events in refuge areas.

**5.2 Climate changes at Lake Ohrid: comparison with independent proxies and other climate**
**reconstructions**

Our data are in agreement with climate signals depicted in pollen and geochemical data from the DEEP site (Francke et al., 2016, Wagner et al., 2017) and other Lake Ohrid cores (core JO2004 from the south of the lake, Bordon, 2008; core Co1202 from the north-east, Holtvoeth et al., 2017; see Fig. 1). When comparing our results to the Eemian climate reconstruction of JO2004 (Bordon, 2008), the trends are
similar, while some differences in temperature and precipitation values should be pointed out. They probably result from differences in pollen assemblages due to the different positions of the analysed cores. Core JO2004 was retrieved from the southern part of the lake, closer to the lake shoreline. Therefore, its pollen assemblages show increased values of local taxa and of those not dispersed over long distances; in contrast, these taxa are found in lower abundance or not at all in the central part of the
basin from where the DEEP core was retrieved. Due to the central position of the DEEP and the morphology of the territory around the lake (vegetation organized in altitudinal belts) we think that our climate reconstruction integrates the palynological signal of the surrounding mountain ranges and, consequently, our data accounts for a regional and not a local climate reconstruction. In Figure 4, the temperature and precipitation (PANN and TANN) signals are compared to the total inorganic carbon
(TIC) and the total organic carbon (TOC) records from the DEEP core and to the TIC and Tetra Ether



inde X of archaeal lipids (TEX$_{86}$) from core Co1202. For more information about these proxies see
Francke et al. (2016), Vogel et al. (2010), Holtvoeth et al. (2017) and references therein. All proxies
reported in Figure 4 are used as indicators for environmental and climatic change. Concerning proxies
from the DEEP core, PANN and TANN resemble TIC and TOC. TIC concentrations and precipitation
of mainly authigenic carbonate is controlled by water temperature and productivity, but also by ion
concentrations in the lake, which depend on precipitation and the activity of the karst aquifer system
(Vogel et al. 2010, Francke et al., 2016). Minima in TOC that correspond to minima in TANN indicate
that these minima are the result of restricted productivity combined with increased decomposition of
organic matter due to the prolonged winter season and enhanced mixing of the water column (Francke
et al., 2016). However, TOC reflects autochthonous and allochthonous organic matter input, i.e. supply
of biomass from both the lake as well as the surrounding land (Francke et al. 2016, Holtvoeth et al.,
2017). The productivity of the terrestrial vegetation and export of terrestrial organic matter seem to be
largely controlled by precipitation rather than temperature, thus, explaining similarities with the PANN
record. While TIC and TOC may co-vary at times they are not generally causally related. During MIS 6,
TIC is mostly very low, suggesting cold and dry climate conditions (Francke at al., 2016), in agreement
with the pollen-inferred mean annual temperature and precipitation (Fig. 4). At the transition toward
MIS 5, TIC and TEX$_{86}$ values increase together, indicating a warming and augmentation of humidity,
consistent with the increase in PANN and TANN inferred from pollen. The distinct high-amplitude
fluctuations inferred from pollen during the final part of MIS 6, could at least partly be due to lake level
changes as the water table during this period was generally on the rise (Lindhorst et al., 2010, Holtvoeth
et al., 2017; Wagner et al., 2017). As mentioned before, the (modern) lake basin and parts of the lake
floor show a pronounced terraced morphology. The relatively rapid flooding of nearly horizontal
surfaces, in particular at the northern and southern ends of the lake, may thus have diminished sizeable
parts of (flat) terrestrial habitat in short periods of time. The impact of lake level change on the low-
lying terrestrial habitats could be clearly seen in the biomarker and pollen records of the proximal
Co1202 (Holtvoeth et al., 2017). While localised processes are likely averaged out by the longer-
distance transport of material towards the distal DEEP, a basin-wide effect of lake-level change and the
associated distribution of low-lying biomes in the north, northeast and south of the basin might have to
be considered in order to explain the observed fluctuations in the pollen record.
The beginning of the Last Interglacial is almost synchronous as indicated by the records of TIC (DEEP),
carbonate and TEX$_{86}$ (Co1202, Figure 4). However, according to the TIC and TEX$_{86}$ records of Co1202
the thermal optimum, characterized by stable conditions and warm temperature, occurs between 126.5
ka and 124 ka, in contrast to our reconstructed temperature that increases earlier, at 127 ka. This slight
discrepancy is probably due to differences in the chronology established for the two cores. An
explanation for the delay of TIC values takes into account the time needed for the dissolution of calcite
from the surrounding rocks (see Francke et al., 2016). At long time scales, calcite precipitation occurs
during periods of high precipitation such as interglacials and interstadials when supply of calcium and
carbonate ions from calcite dissolution into the lake increases, and/or elevated temperature and high
evaporation occur. Biogenic calcite formation is hampered during dry and cold periods (glacial and
stadials) due to decreased precipitation and associated nutrient supply and reduced terrestrial calcite
dissolution and inflow of dissolved carbonate from the karst system (Lézine et al., 2010). After the
"climate optimum" (128-121 ka), from 120 ka and culminating at 119.4 ka, TIC values decrease



together with mean annual temperature, by contrast, in the same time precipitation rises. The low TIC
content can be explained by lower water temperature, which hampers calcite precipitation. Progressive
drying occurs from 121 ka until the end of Eemian at 112 ka. This trend is more or less followed by TIC
and the pollen-inferred climate trends at Lake Ohrid, corroborating the climate reconstruction based on
core JO2004 (Bordon, 2008). This trend also confirms the assumption that the Last Interglacial was not
a uniform wet and warm phase in western Europe (e.g. Cheddadi et al., 1998; Guiot et al., 1993; Klotz
et al., 2004; Kühl and Litt, 2003; Rousseau et al., 2006; Sánchez-Goñi et al., 2005) and that successive
cool/dry events occur at ca. 110 and 105 ka.

### 5.3 Comparison with European climate reconstructions inferred from pollen records

Lake Ohrid's chronology is well established for MIS 5 due to the high number of tephra layers (Francke
et al., 2016; Leicher et al., 2016), in particular, for the transition between the Riss glaciation and the
Eemian, for which a further correlation with geo-chemical and pollen data from Lake Ohrid and other
proxies from Mediterranean sequences was carried out by Zanchetta et al. (2016). For other European
pollen records such chronological constraints are not available and, thus, the chronologies are less
precise. Keeping in mind the existing chronological uncertainties, a comparison of precipitation and
temperature anomalies is carried out, with the values inferred from three other long pollen records (Fig.
5) spanning the interval between 140 and 70 ka: Les Echets, Le Bouchet and La Grande Pile (Fig. 1).
Lake Ohrid, despite being considered as "a southern site", shows past climate trends similar to the
French records (Fig. 5). This similarity is probably due to its high elevation, causing enhanced
precipitation in relative to the rest of southern Europe and making it similar to regions directly subjected
to the North-Atlantic circulation. In order to discuss Lake Ohrid's climate record more in depth on a
European scale, a further comparison is shown in Figure 6. Here, Lake Ohrid climate anomalies are
plotted with the ones estimated by Brewer et al. (2008) for southern and central-northern European
sites, using a pollen–inferred multi-method approach which take into account the various sources of
errors in paleoclimate reconstructions. The investigated interval is in this case limited to the period 135-
105 ka, which includes the whole Eemian (ca. MIS 5e-d according to Sánchez-Goñi et al., 2007).
During the final part of MIS 6 (Fig. 3, 6), climate seems to have been particularly harsh at Lake Ohrid,
with highly reduced precipitation both compared to other European sites (Brewer et al., 2008) or to
present. However, the precipitation anomaly values are comparable to those of the French sites (Fig. 5).
30   For the latter, we have to consider that the same methods have been applied, which could have resulted
in the more consistent values. La Grande Pile, Les Echets and Le Bouchet reconstructions show a
"climate optimum" from 127 to 118 ka followed by an abrupt cooling around 117 ka (fig. 6 and Brewer
et al., 2008). The signal reconstructed for northern Europe is different from the French sites, Brewer et
al. (2008) had identified a climate tri-partition during the Eemian: early optimum, followed by slight
35   cooling, followed by a sharp drop in temperatures and precipitation. This set of changes appears
restricted to the north, with a very different set of changes in south. In southern Europe, the Eemian
climate appears to have remained warm with stable conditions over a long period between 126-105 ka
(Fig. 6). Lake Ohrid is located in a central position on the Balkans Peninsula, at the confluence of
central European and Mediterranean climate. The Lake Ohrid climate reconstruction also shows a
40   "climate optimum" in the early part of the Eemian and then a progressive cooling without a sharp drop

in temperatures and precipitation (Fig. 6); this suggests an intermediate climate signal, more similar to the French sites (Fig. 5) than to the northern or southern European ones (Fig. 6). Brewer et al. (2008) show that climate changes during this period were heterogeneous, with greater winter warming in the center and north-east of Europe than in the west and north-west. Other studies of the spatial distribution

of temperature changes during this period have shown similar trends in temperature, with the largest positive anomalies in central and northern Europe, and negative anomalies in south-eastern Europe (Kaspar et al., 2005; Turney and Jones, 2010; Otto-Bliesner et al., 2013). Furthermore, one remaining question is whether the climate of this period was very close to modern values or warmer and wetter than the present-day as suggested by existing studies (Guiot et al., 1989). The time series of anomalies

presented here (Fig. 5) suggest a positive anomaly of 1 to 2°C for the Ohrid Basin, strongly depending on the method used (Fig. 5). Melisey I is the first cooling event, with a significant reduction in temperatures and precipitation, although less pronounced than at the French sites (Fig. 5). At Lake Ohrid, a surprising positive anomaly in the middle of Melisey I is suggested and is potentially due to the persistence of trees during stadials, highlighting the important role of the Ohrid Basin as a refugium for

arboreal taxa. According to several studies carried out in central and northern Europe (Guiot et al., 1993; Klotz et al., 2004; Rioual et al., 2001), the Melisey I event is characterized by an abrupt decline in temperatures first, followed by increasing continental conditions, with a subsequent decline in winter temperatures and an increase in summer temperatures. Other pollen records from Lake Ohrid also strongly suggest that climatic conditions remained favorable to grow mesophilous taxa (Bordon, 2008;

Holtvoeth et al., 2017; Lézine et al., 2010). St. Germain Ia (Figs. 3 and 5) is drier than St. Germain Ic at Lake Ohrid, with the latter showing annual precipitation up to ca. 400 mm/yr higher than during the former. The values are consistent with the data obtained by Klotz et al. (2004) for central Europe, more specifically, in the northern Alpine foreland. The same trend is also recorded in the French sites presented here (Fig. 5). Melisey II appears as the most extreme stadial of the LIC, coinciding with the

maximum extension of ice sheets during the Early Weichselian. However, the cooling reconstructed at Ohrid is probably overestimated with the MAT for the same reasons as during MIS 6. If we consider the WAPLS reconstruction, the anomalies estimated at Ohrid during Melisey II are 2 °C higher than for the French sites (Figs 3 and 5). During St. Germain II, temperature and precipitation values for Lake Ohrid are similar to those of St. Germain Ia (Figs 3 and 5). This pattern is corroborated by other studies for the

North Atlantic, using marine $\delta^{18}O$ data (Keigwin et al., 1994), for North Europe (e.g., Guiot et al., 1989) and for the Iberian Margin (Sánchez-Goñi et al., 2000). At the end of the interstadial, a trend towards low temperatures and an increase in precipitation is recorded at Lake Ohrid, in agreement with the climate reconstruction of Guiot et al. (1989) for the French pollen records (Fig. 5). The most striking feature of Lake Ohrid, recorded at the top of the studied sequence, is the presence of two interstadials

following St. Germain II, namely Ognon I and Ognon II.

### 5.4 Comparison with other European and North Atlantic proxy records

In order to discuss the Ohrid climate signal at a wider scale, Figure 7 shows the correlation of the reconstructed climate parameters with marine and continental proxies from Mediterranean and North Atlantic regions (Fig. 1).




In speleothem and lake sediment records, $\delta^{18}O$ is mostly seen as an indicator of the "amount of precipitation", lower/higher values are related to increasing/decreasing humidity (Bard et al., 2002; Drysdale et al., 2005, 2009; Regattieri et al., 2014; Zanchetta et al., 2007, 2016). The Ohrid precipitation trend shows similarities with the oxygen isotope records reported in Figure 7, suggesting a

generally good agreement with the variations in Mediterranean rainfall detected in Italy in speleothems from Antro del Corchia and Tana che Urla (Drysdale et al., 2005; Regattieri et al., 2014) and in the lake record of Sulmona (Regattieri et al., 2017). According to Drysdale et al. (2009), there is a pause in the decrease in $\delta^{18}O$ continental and marine values prior to the beginning of the Eemian at ca. 129 ka, which can be related to Heinrich event 11 (H11, Shackleton et al., 2003). During this event, the North

Atlantic thermohaline circulation and the North Atlantic deep-water formation shut down with a consequent phase of cooler and drier conditions for mid-latitude Western Europe (Genty et al., 2003). At Lake Ohrid (Fig. 7), H11 is clearly detected, for the first time in a climate reconstruction, and in the TIC records of the DEEP core and Co1202 core (Figs 4, 7).
The climatic changes evidenced at Lake Ohrid during the Eemian (at ca. 126 ka, 122 ka, 119 ka and 114

ka, Fig. 7) are particularly consistent with speleothems and $\delta^{18}O$ Sulmona records (see Fig. 7). Important changes during the LI have also been detected in the alkenone-based sea surface temperature (SST) reconstruction of the ODP-976 and ODP-977 sediment cores (Alboran Basin, Martrat et al., 2014), in $\delta^{18}O$ records of the Iberian Margin (MD95-2042 and MD95-2040, Sánchez Goñi et al., 1999, 2005; De Abreu et al., 2003), Villars cave (Wainer et al., 2011) and from Greenland (Fig. 7), in line

with other studies on speleothems and on Mediterranean and North Atlantic marine records (e.g. De Abreu et al., 2003; Drysdale et al., 2009; Lisiecki and Raymo, 2005; Martrat et al., 2007, 2014; McManus, et al., 1994; Mokeddem et al., 2014; NGRIP, 2004; Oppo et al., 2006; Sánchez Goñi et al., 1999, 2005; Wang et al., 2010). Based on the ODP-976 and ODP-977 alkenone data (Martrat et al., 2004, 2014), warm SSTs occurred during interstadial periods, while cold SSTs persisted during stadials

Melisey I and II. SST changes are associated to large shifts in mean annual air temperature and moisture content as reflected in vegetation changes inferred from pollen analysis in European and Mediterranean records (Martrat et al., 2014; Tzedakis et al., 2003). The event centered at ca. 115 ka is of particular interest (Fig. 7). According to Müller and Kukla (2004), this SST drop in the Nordic Seas marked a southward displacement of the North Atlantic Current. This event was associated with a substantial

cooling in North Europe and drier conditions in Mediterranean regions, marking the end of the Eemian in northern and central Europe (Müller and Kukla, 2004; Tzedakis et al., 2003), while in the south it correlated with a reduction in mesophilous trees. This event is also recorded at Lake Ohrid (Fig. 7) with a simultaneous drop in temperature and precipitation values, and is associated to the GIS26 stadial of the Greenland ice core record (NGRIP, 2004) and to the C25 cold event in the North Atlantic

(McManus et al., 1994; Mokeddem et al., 2014). This connection between Lake Ohrid and the North Atlantic (Fig. 7) is also highlighted by the evidence of the Melisey I stadial, which corresponds to North Atlantic event C24 (and to GS25), the Montaigu event, corresponding to C23 (and GS24), and the Melisey II stadial, which corresponds to C21 (and GS22). Besides this event, the final part of MIS 5 at the transition with MIS 4 at Lake Ohrid is characterized by a series of abrupt climate changes (Ognon I

and II phases), composed of two interstadials and two stadials. The latter correspond to the North Atlantic cold events C20 (GS21) and C19 (GS20), respectively (Fig. 7). A similar pattern can be





depicted in the SST record of ODP-977 (Fig. 7), with two abrupt warming events, preceded by a strong cooling after a long period of stability (Martrat et al., 2004).

## 6 Conclusion

The temperature reconstructions from both methods are reasonably coherent (trends and values) during the interglacial and We provide a quantitative reconstruction of climate parameters based on the pollen record from Lake Ohrid (DEEP site), using two complementary approaches for the period between 160 and 70 ka. This period covers the last part of the Riss Glaciation, equivalent to MIS 6 (160-128 ka), and the Last Interglacial Complex (128 to 70 ka), equivalent to MIS 5, as well as the first part of MIS 4. Our results for the LIC show an alternation of warm and wet periods (128-112 ka, 108-89.8 ka, 85.7-78.8 ka, 77.6-75.4 ka, 73.4-70 ka) with cold and dry ones (112-108 ka, 105.2-104 ka, 89.8-85.7 ka, 78.8-77.6 ka and 75.4-73.4 ka) attributable to the well-known succession of climatic events occurring during MIS 6 and 5.

With regard to the last interglacial, our results provide evidence that the Eemian was not as stable as previously thought, confirming existing studies. The climate reconstruction led to distinguish three periods: a climatic optimum (128-121 ka), followed by progressive cooling in conjunction with an increase in precipitation (121-118 ka), and, finally, a period characterized by a decrease in both temperatures and precipitation (118-112 ka).

The early last glacial (from 112 to 70 ka) is characterized by a succession of cold and warm periods (stadials and interstadials) in which cold ones show an increase in seasonality and dry conditions. This climatic trend can be correlated to the succession of Greenland stadials and of North Atlantic cold events (Dansgaard et al., 1993; GRIP Members, 1993), illustrating the teleconnections between the North Atlantic realm and the Mediterranean region. The same succession of cold and dry events at Lake Ohrid is also coherent with hydrological and isotopic data from the central Mediterranean.

At a wider scale, our results showed a great similarity between Lake Ohrid and climate reconstructions of French and central European records rather than the stacked curve of four southern European records. Lake Ohrid shows intermediate features between these two areas; our curves are in line with those of other southern European climate proxies (e.g. central Italian speleothems). Future climate reconstructions and independent proxies are needed for the southern Mediterranean to resolve the complex regional expressions of past climate changes.

## Author contributions

The paper was written by GS (all sections) and LS (Sects. 1, 2, 4, 5, 6) with the substantial contribution of AF (Sects. 4, 5, 6), AM (Sects. 3, 5, 6), BW (Sects. 1, 4, 5, 6), JH (Sects. 1, 4, 5, 6) and OP (Sects. 3, 5, 6). Pollen analysis was carried out by GS with the contribution of AM and LS. AM is responsible for pollen data management and the elaboration of figures and diagrams. The quantitative reconstruction of temperature and precipitation was carried out by GS under the guide and supervision of OP.





**Acknowledgments**

This work was developed in the frame of the joint Ph.D. in Earth Sciences between the University of
Rome "La Sapienza" (XXX cycle) and the University of Montpellier of Gaia Sinopoli. Moreover, G. S.
acknowledges the Vinci program, which granted her mobility between the two universities. This is an
ISEM contribution.
The authors thank the palynological team involved in the SCOPSCO project. The drilling of the DEEP
core was funded by ICDP, the German Ministry of Higher Education and Research, the German
Research Foundation (DFG), the University of Cologne, the British Geological Survey, the INGV and
CNR of Italy, and the governments of the Republic of Macedonia (FYROM) and Albania. Logistic
support was provided by the Hydrobiological Institute in Ohrid. Drilling was carried out by Drilling,
Observation and Sampling of the Earth's Continental Crust's (DOSECC) and using the Deep Lake
Drilling System (DLDS).

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



**Table**

| MARINE STRATIGRAPHY | COMMON NAME | LA GRANDE PILE |
|---|---|---|
| MIS 4 | Early Last Glacial | Ognon II Stadial II Ognon I Stadial I St. Germain II |
| MIS 5 | | Melisey II |
| | | St. Germain Ic St. Germain Ib (Montaigu event) St. Germain Ia |
| | | Melisey I |
| | Last Interglacial | Eemian |
| MIS 6 | Riss Glaciation | |

5    **Table 1 Correlation of nomenclature defined by Woillard (1978) for La Grande Pile (NE France) with common terrestrial nomenclature and the Marine Isotope Stages (MIS, Lisiecki and Raymo, 2005).**





**Figure**

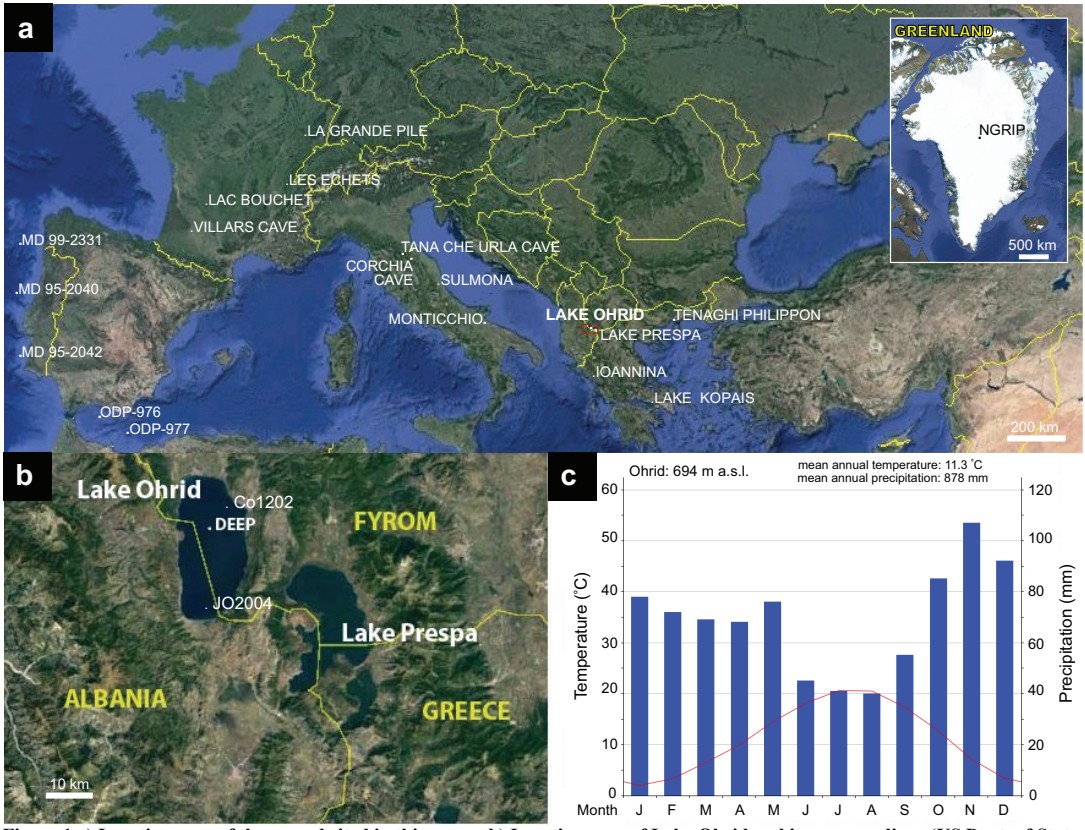

**Figure 1 a) Location map of the record cited in this paper. b) Location map of Lake Ohrid and its surroundings (US Dept. of State Geographer © 2017 Google Image Landstat/ Copernicus, Data SIO, NOAA, U.S. Navy, NGA, GEBCO) c) Ombrothermic diagram of Struga metereological station (http://en.climate-data.org/location/29778/).**



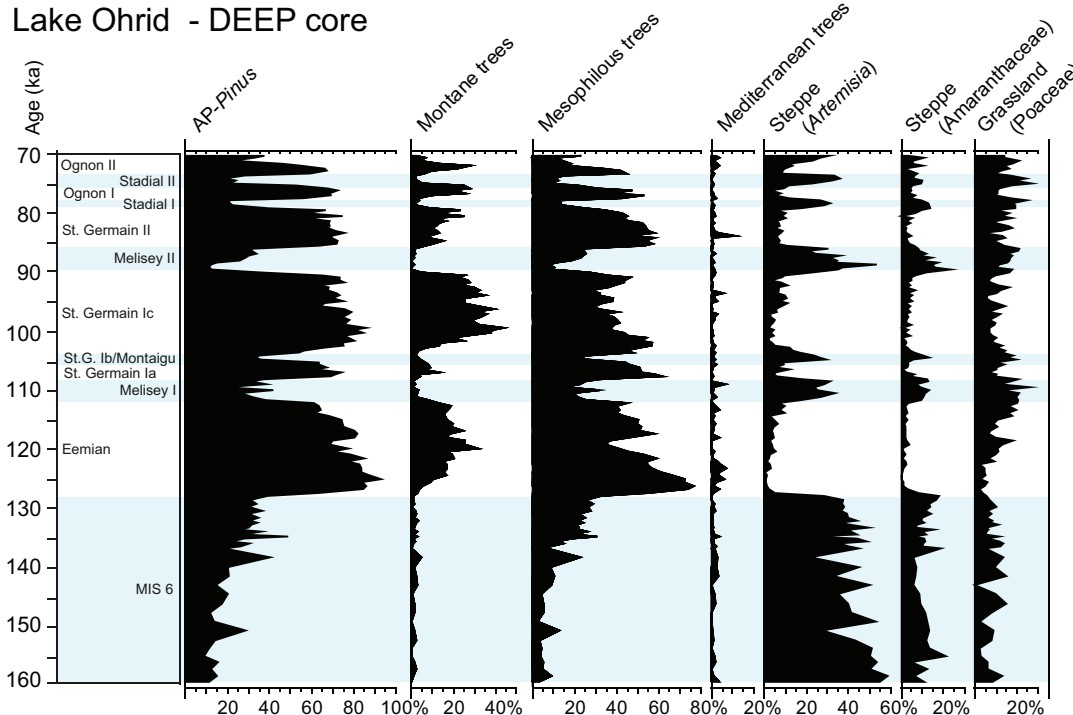

**Figure 2 Lake Ohrid (FYROM/ Albania) - DEEP core - Pollen percentage diagram of selected taxa and ecological groups, against
age (ka). Montane trees:** *Abies, Betula, Fagus, Ilex, Picea*; **mesophilous trees:** *Acer, Buxus, Carpinus betulus, Castanea, Celtis,
Corylus, Fraxinus excelsior/oxycarpa, Ostrya/Carpinus orientalis, Hedera, Quercus robur* **type,** *Quercus cerris* **type,** *Tilia, Ulmus,
Zelkova*; **mediterranean trees:** *Cistus, Fraxinus ornus, Olea, Phillyrea, Pistacia, Quercus ilex* **type,** *Rhamnus*; **steppe: Artemisia,
Amaranthaceae, Chicoriodeae and Asteroideae) and grassland: Poaceae and Cyperaceae. Data from Sadori et al. (2016) and
Sinopoli et al. (2018).**



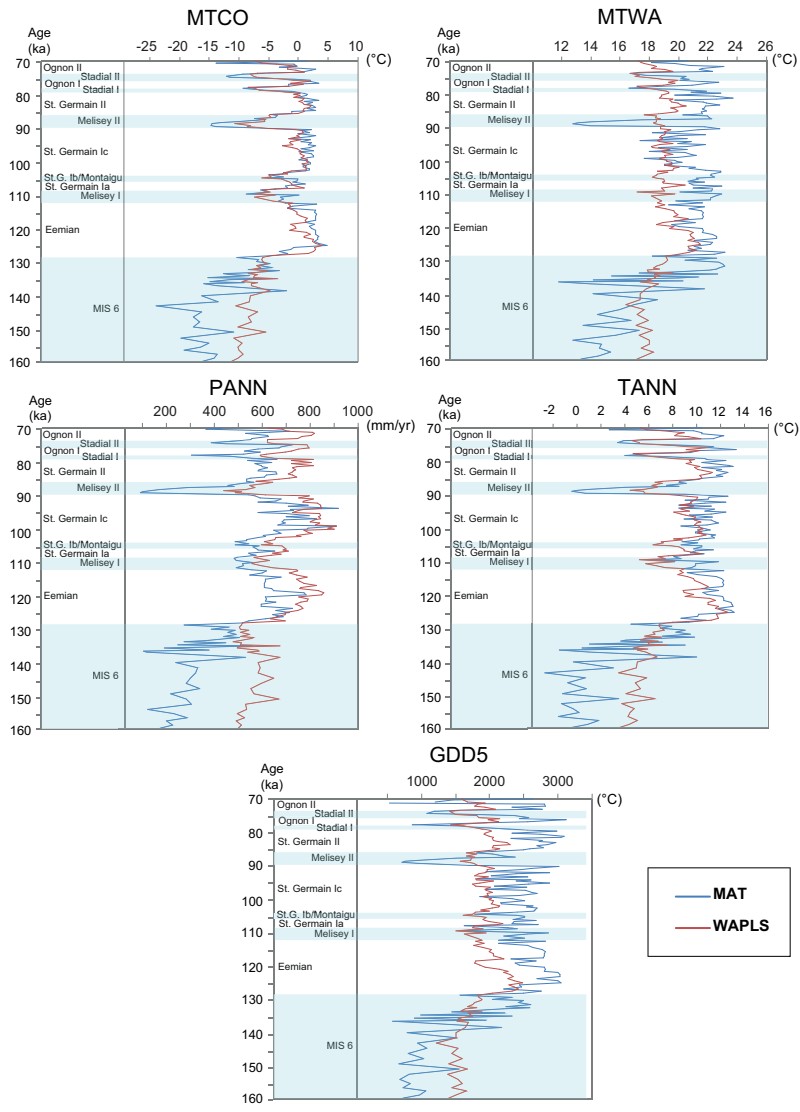

**Figure 3 Climate reconstruction inferred from Lake Ohrid pollen data (Sadori et al., 2016; Sinopoli et al., 2018). Climate parameters obtained with the MAT method (black line) and the WAPLS method (red line): MTCO (mean temperature of the coldest month), MTWA (mean temperature of the warmest month), PANN (mean annual precipitation), TANN (mean annual temperature) and GDD5 (growing degrees days over 5 °C). Climate parameter values are plotted against age (ka); they are not expressed in anomalies (past climate value minus the modern measured value). Blue shading indicates cold periods (Riss glacial and Early Würm glacial stadials). Blue shading indicates cold periods (Riss glacial and Early Würmian glacial stadials).**





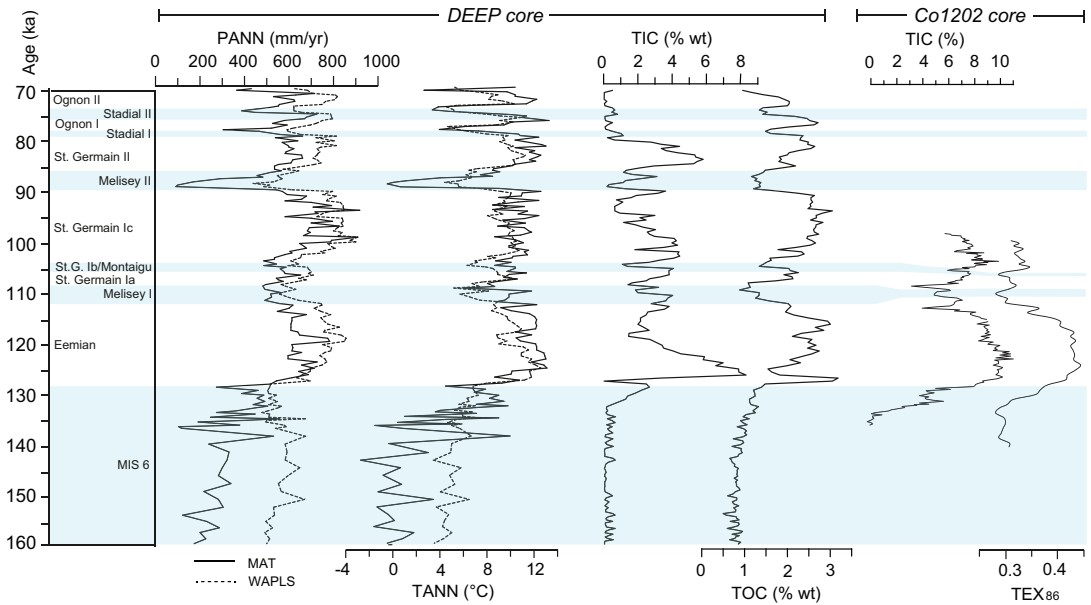

**Figure 4 Lake Ohrid: Comparison between DEEP core annual temperature (TANN), precipitations (PANN) and TIC (Francke et al., 2016) with TIC and TEX86 (Co1202 core, Holtvoeth et al., 2017). Blue shading indicates cold periods (Riss glacial and early Würm stadials).**





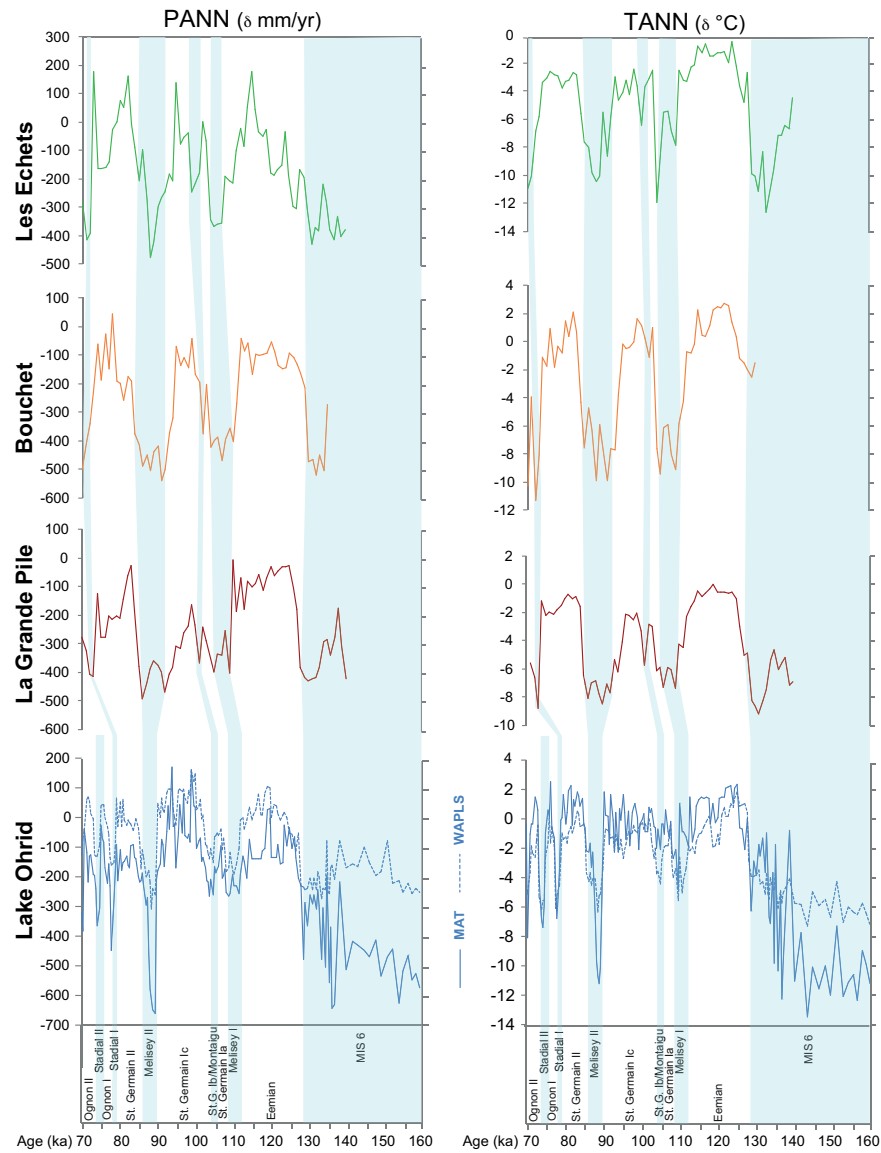

**Figure 5 Comparison between Lake Ohrid climate parameters with available climate reconstructions: Les Echets (265 m a.s.l.), Le Bouchet (1200 m a.s.l.) and La Grande Pile (330 m a.s.l.) from Guiot et al. (1989, 1990, 1993). TANN (mean annual temperature) and PANN (mean annual precipitation) are plotted against age (ka). Values represent anomalies (past climate value minus the modern measured value). Blue shading indicates cold periods (Riss glacial and stadials).**





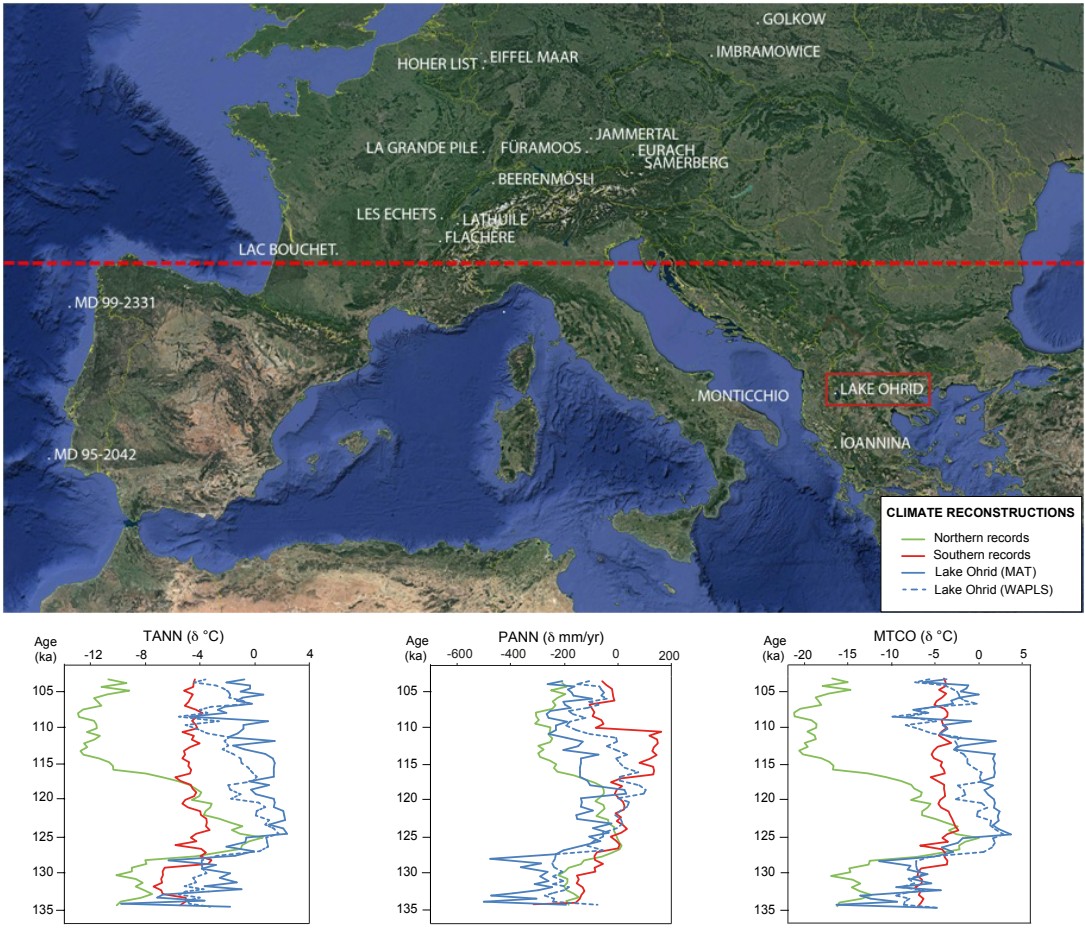

Figure 6 Comparison between the Lake Ohrid climate reconstruction and the climate reconstruction performed by Brewer et al., (2008) for North and South Europe; TANN (mean annual temperature), PREC (mean annual precipitation) and MTCO (mean temperature of the coldest month) are plotted against chronology (ka). Values represent anomalies (past climate value minus the modern measured value).





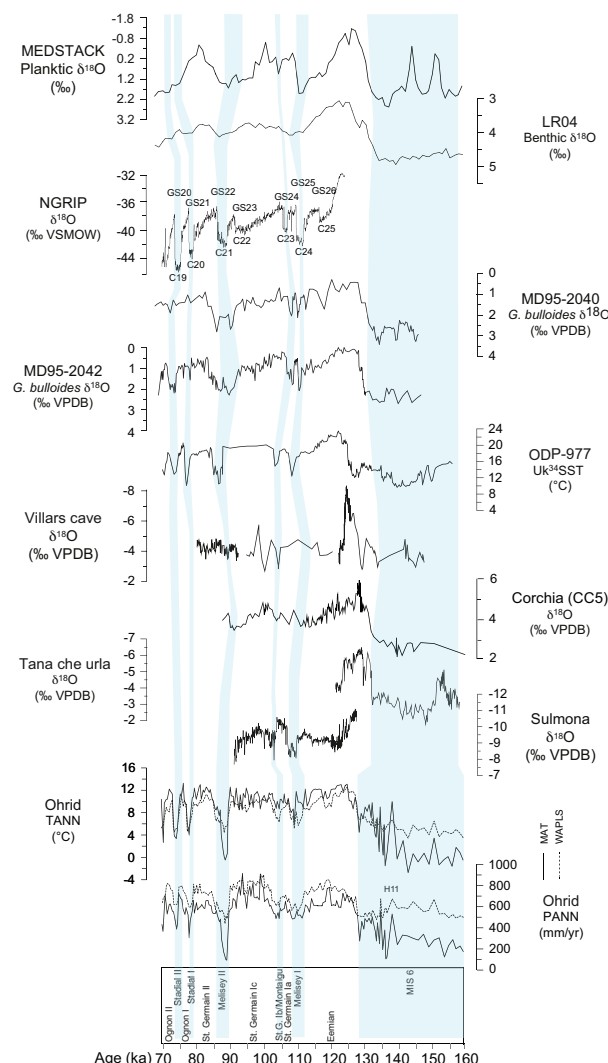

**Figure 7 Comparison of TANN, PANN values from Lake Ohrid with other hydrological and climate proxies from the Mediterranean and the North Atlantic: planktic δ¹⁸O from MEDSTACK data (Wang et al., 2010); δ¹⁸O benthic from LR04 stack (Lisiecki and Raymo, 2005); Greenland δ¹⁸O record (NGRIP, 2004); planktic δ¹⁸O from Iberian Margin (MD95-2040, De Abreu et al., 2003); planktic δ¹⁸O from Iberian Margin (MD95-2042, Sánchez Goñi et al., 1999; Sánchez Goñi et al., 2005); Sea Surface Temperature (SST) from core ODP-977 (Western Mediterranean, Martrat et al., 2004); δ¹⁸O speleothem record from Villars cave (Vil-car-1, Wainer et al., 2011); δ¹⁸O speleothem record from Corchia cave (CC5, Drysdale et al., 2005); δ¹⁸O speleothem record from Tana che Urla Cave and Regattieri et al., 2017; δ¹⁸O of endogenic calcite from Sulmona Lake (Regattieri et al., 2014). Numbers denote Greenland Stadials (GS), corresponding to North Atlantic cold events (C events, after McManus et al., 1994). Blue shading indicates cold periods (Riss glacial and stadials).**



**Appendix**

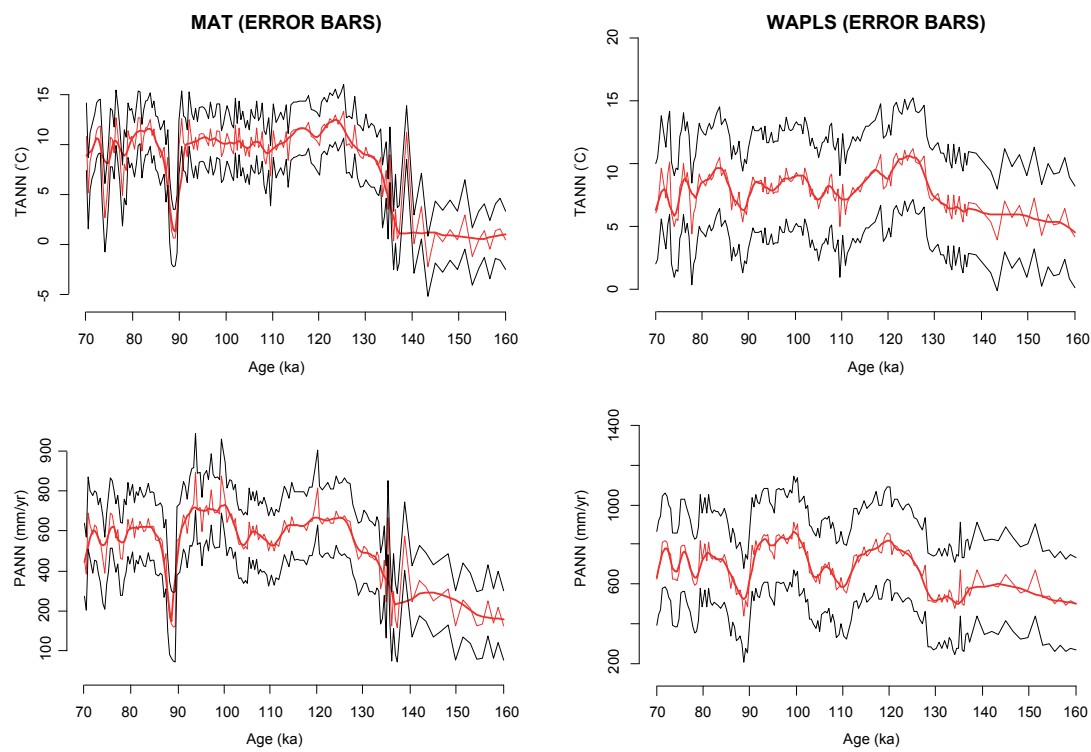

**Figure A1 MAT and WAPLS error bars for PANN and TANN climate parameters. The last graph represents the squared-chord distance between the first and the last analogue for a chosen climate parameter (TANN) calculated by MAT method.**



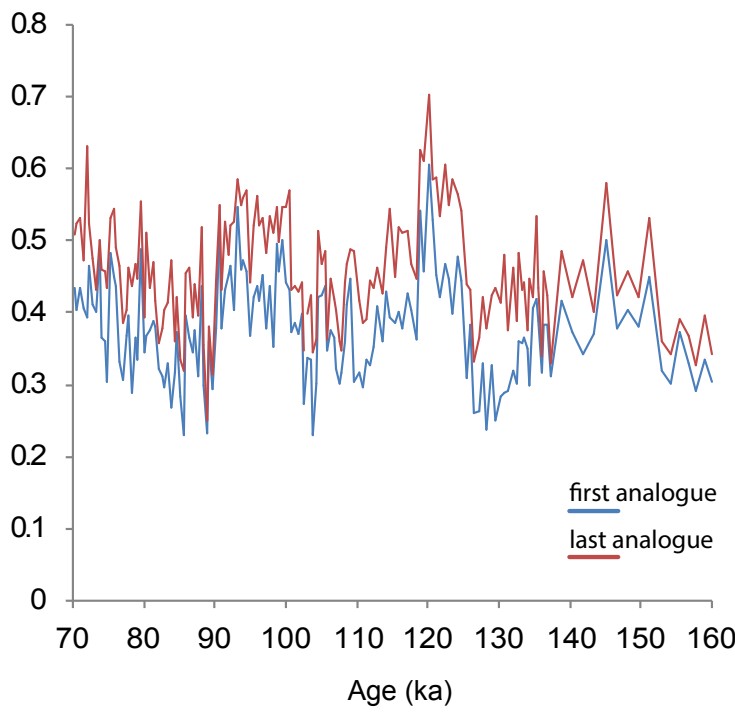

**Figure A2 Squared-chord distance between the first and the last analogue for a chosen climate parameter (TANN) calculated by MAT method.**