# Peer review of "Pollen-based temperature and precipitation changes in the Ohrid Basin (western Balkans) between 160 and 70 ka"

_Climate of the Past, 2018_

## Short Comment (SC1) · 10 Jul 2018

Dear Authors,

Congratulations for this interesting paper, I hope it'll go through. Regarding the comparison of the last interglacial section with speleothem data, I would like to call your attention to our paper on a Hungarian speleothem (Demény et al., 2017, Stable isotope compositions of speleothems from the last interglacial – Spatial patterns of climate fluctuations in Europe. QUATERNARY SCIENCE REVIEWS 161: pp. 68-80.). The d13C and d18O data of the BAR-II speleothem show very good agreements with your precipitation and temperature reconstructions. I pasted our plot onto your Fig 7

(sorry for the preliminary outlook). The d13C data show a negative peak (inverted scale!) at about 118 ka when you have a precipitation peak. The d18O values indicate dry and still relatively warm conditions (high evaporation) around 117 ka, when your precipitation reconstruction shows a decrease while the temperature increases. I think the good matches support your conclusion about the Central European meteorological connections. I hope your can use these data in the revision.

Best regards, Attila Demény
* * *
[Figure]

**Fig. 1.** The BAR-II records with Fig 7

---

## Short Comment (SC2) · 18 Jul 2018

Dear Attila, thank you very much, we are grateful for your suggestion and we will take it in consideration.

With kind regards Alessia

---

## Referee Comment (RC1) · Anonymous Referee #1 · 5 Sep 2018

This is a very interesting paper worthy to be published and here are my suggestions for some improvements following the continuity of the text.

P1, Line 25 (and p.14 lines 13-17). For me, the discussion on the stability of the Eemien rests on a misdefinition. Since Jessen, everybody know that an interglacial cycle includes a period of warming after the previous glacial, an optimum and finally a progressive cooling leading to the next glacial. In this way the instability you mention is a truism! You could just mention that "The Eemian in the Balkans was characterized by an abrupt early warming during its anathermic phase followed a central phase . . ..). Most of the authors, when dealing with "instability" or "stability" try to identify some

short terms climate "oscillations" disturbing the classical interglacial trend on a warming followed by a cooling (as well discussed latter in your paper).

P2, l.30-35. Before Field et al. (1994), at least Beaulieu & Reille (1984,1889) already mentioned a period of transitional warming during the late Eemian.

P3, l.2 and fig. 6: be careful with the coordinates. Les Echets: 45°52'36" N, 4°55'44" E and Lac du Bouchet : 44°54'31" N , 3°47'30". Lac du Bouchet is really transitional between North and South according to your classification. As a matter of fact, this limit at 45°N is of interest as far as the Eemian is concerned, but during the Early Würm the story is more complex with the opposition between the "Odderade style" vegetation and climate successions and the "Grande Pile style" successions. It could be mentioned.

P3, l.27. : I should prefer "higher resolution", as an interval of 400 is not high resolution in terms of vegetation dynamics. Would you accept a pollen diagram covering the whole Holocene in only 28 spectra?

P4, the first sentence is not necessary as the following description is sufficient.

P4, l.11: do not repeat "karst aquifers".

P5, top: my copy is polluted by three lines in pseudo-latin.

P5, l.34: could you develop and explain in more details your choice of 6 modern analogues?

P6, l. 20:I suggest "pollen records " instead of "data"

P6, l.35: again one line polluted with latin.

P9, l.5 and after: this interesting discussion could be included in the chapter "Materials and methods"?

P9, l.22: may-be a clumsy statement. If your climate reconstructions are derived from pollen spectra, it would be a great disaster for your results if they were not in agreement

with their sources.

P10, l. 24-29: Very interesting but I do not understand how the discussion is inferred from fig. 4: TIC and TOC do not change (slight increase in TOC) during the interval between 137 and 135 Ka marked by high amplitude changes in PAN and TAN??

P10, l 34 : Not that slight??

Chapter 5.4. : it would be of interest to take into account the climate reconstructions based on diatoms populations established by Rioual et al. (2007) at Ribains ( see also Shemesh et al., 2001)

References

Beaulieu J.-L. de & Reille M., 1984. A long upper Pleistocene pollen record from Les Echets near Lyon, France, Boreas, 13, p.111-132.

Beaulieu J.-L. de & Reille M., 1989. The transition from temperate phases to stadials in the long Upper Pleistocene sequence from les Echets (France). Palaeogeography, ,Palaeoclimatology, Palaeoecology, 72, 147-159.

Rioual P.., Andrieu-Ponel V., Beaulieu J.-L. de, Reille M., Svobodova H. & Battarbee R. W., 2007. Diatom responses to limnological and climatic changes at Ribains maar (French Massif Central) during the Eemian and Early Würm. Quaternary Science Reviews, 26 (11-12), 1557-1609.

Shemesh A., Rietti-Shati M., Rioual P, Battarbee R., Beaulieu J.-L. de, Reille M. and Svobodova H., 2001. An Oxygen isotope record of lacustrine opal from a european Maar indicates climatic stability during the last interglacial. Geophysical Research Letters, 28 (12), 2305-2308.

---

## Referee Comment (RC2) · Anonymous Referee #2 · 22 Sep 2018

Thank you for the opportunity to comment on this manuscript. The manuscript titled 'Pollen-based temperature and precipitation changes in the Ohrid Basin (western Balkans) between 160 and 70 ka' covers a sound data set of great value to the palaeoecological community as climate reconstruction for the last interglacial-glacial cycle from this sensitive area are scarce. Overall, I think the work is good and should be published in CP but there are a number of important details that need to be considered and corrected first. In many instances these are related to terminology, definition of terms and ambiguity or circularity in the phrasing. One important example of this is the use of different nomenclatures, e.g., from alpine region (Riss/Würm glacial), from northern/central Europe (Eemian interglacial), and the special nomenclature from France

for interstadials/stadials. Regarding the last interglacial, be careful with the statement that the Eemian was not a stable phase in the Balkan region. The last interglacial at LO clearly shows a classical interglacial with an early warming at the beginning of MIS 5e, a climate optimum and a progressive cooling towards the end of the last interglacial. In general, I suggest to interpret the LO record with regard to further regional climate reconstructions (mentioned in the chapter 'Introduction', e.g., LGdM and Ioannina) and use it as a basis for discussing in more global scale with possible correlations to the France, speleothem records, MIS, etc. These important issues and more are detailed below along with some suggestions for grammatical corrections.

Page, Line. Comment

P1, 17. The presented archive covers the period between 160 to 70ka. This includes not only MIS 6 and MIS 5, but also the early part of MIS 4. The authors mentioned it in the conclusion by themselves. (P14, 8).

P1, 25-28. According to the anonymous referee #1. The last interglacial at Lake Ohrid shows a classical interglacial cycle with pre-temperate phase (early warming), temperate phase (climate optimum), and a post-temperate phase (progressive cooling), which is also well described by Tzedakis, 2007. Be careful with this general conclusion of an unstable last interglacial in the Balkan region!

P2, 3. Insert '…'MIS' – 6 (penultimate glacial) and MIS 5 (last interglacial complex) are….'

P2, 6. '…the penultimate glacial (or Riss Glaciation) …' in comparison to P2, 12. 'The Eemian…' Please pay more attention to a uniform nomenclature. The term 'Riss glaciation' is normally used in the alpine area. In northern and central Europe, the penultimate glacial belongs to the 'Late Saale/Saalian Complex'. If you want to continue with the term 'Eemian' for the last interglacial (MIS 5e), you should use the nomenclature of the northern and central Europe. Another example is the '(Early) Weichselian glacial' instead of 'Early Würm' (P2, 34).

P2, 7. Please pay more attention to uniformity. The LIC lasts from ca. 130-80 ka in the chapter 'Introduction', whereas the LIC covers the period from ca. 128-70 ka at page 3, 25. Please check the dates, there are several more discrepancies.

P3, 2-4. The authors mentioned that Ioannina and LGdM have done climate reconstructions based on pollen data. Unfortunately, these two archives were not used for comparison (e.g. in Figure 5) or were not discussed in detail in the text (chapter 5.3), although these records are much closer to Lake Ohrid than the archives in France. Due to the fact that you mentioned in P3, line 15 that the Balkan Peninsula is a key region between the Mediterranean area and the Northern/Central Europe. It would be nice to see how these few southern European records differ from the northern European ones. What about the direct comparison with Lake Prespa, which covers the last ∼90 ka. I am not sure if they have done climate reconstructions, but are there any similarities or differences to your record?

P3, 14. . . .glacial-interglacial cycle. (?)

P4, 1-2. The first sentence is not necessary.

P4, 11. Avoid the repetition of 'karst aquifers' at the end of line 11.

P4, 12-13. Rephrase: '. . .small streams, rivers, and by direct precipitation.'

P4, 18. Rephrase: '. . .during winter and south-southeasterly (or southerly to southeasterly) winds during. . ..'

P4, 21. This context is not clear – please rephrase. What are the four zones? Which species dominate which zone?

P5, 1-3. Please check. This sentence is written in a different language.

P5, 15. What is 'new' in the high-resolution pollen data, presented in this manuscript, when it is already published in Sinopoli et al., 2018? Did you analyse more 'new' samples for this manuscript, which are not shown in the Sinopoli et al. paper? Please

clarify!

P5, 34. Which 'six modern analogues'? This subject should be further explained and clarified in the text.

P6, 18. It is not clear to me what do you mean with the '. . .first analogue and the last analogue. . .'? More details are needed.

P6, 35-36. Please check. This sentence is written in a different language.

P7, 1. '. . .and annual precipitation between 350 and 600 mm/yr),. . .' It depends on what method you looking at. For MAT, I can recognize a fluctuation from 100 to ca. 300 mm/yr in the mean annual precipitation. For WALPS, it fluctuates between 500 to 700 mm/yr. What is the explanation for this huge difference? Please clarify and add some more explanations.

P7, 11 and 14. Which 'other methods'? If necessary, add references.

P7, 34-36. Which is the third part? Furthermore, an additional verb is missing. Please rephrase this sentence.

P8, 17 and following. Describe the 'end of MIS 6' within chapter 4.1

P9, 8 -16. This section should be mentioned in the chapter 'Materials & methods'.

P9, 19-20. To be consistent with the text, could you add the discussed pioneer shrubs (e.g. Juniperus) to the selected pollen diagram (Figure 2).

P9, 28 and following. For the better understanding and demonstration, it would be very helpful to show the comparison of your Eemian climate reconstructions with those from the JO2004 record. Please insert the JO2004 climate reconstruction, for example, in Figure 4.

P10, 12. The phrase '. . .export of terrestrial organic matter. . .' sounds a bit odd. Please rephrase.

P10, 18 and 19. Delete '. . .inferred from pollen.' due to the repetition from the previous sentence. At this point, it is obvious that TANN and PANN were calculated from pollen.

P10, 18-29. There seem to be some logical steps missing. I cannot work out how lake level changes can be visible in the pollen record. I also cannot see a decline in terrestrial vegetation at the end of MIS 6 - in fact, quite the opposite. It shows a continuous increase in mesophilous and coniferous trees! In addition, I assume that significant lake level changes should be reflected in the TIC /TOC values, but again I cannot see any changes in these proxies at the DEEP site. Furthermore, the 'clearly seen' change in the pollen record of Co2012 is not presented in this manuscript! These subjects should be explained more clearly in the text.

P10, 31-34. In my opinion, a difference of 500 years is NOT a discrepancy. The authors should soften the language.

P10, 39-41. Please avoid the use of so many 'and' in this sentence. Please rephrase.

P10, 42 and following. There is something odd about the line of reasoning here. It is not clear to me what do the authors mean with 'from 120 ka and culminating at 119.4 ka'? In Fig. 4, I cannot identify a 'culmination' in the TIC decrease during this interval. At the DEEP site, the TIC and TOC values already continuously decrease after ca. 126 ka! In addition, how can a progressive drying (P11, 3) take place when precipitation increases at the same time (P11, 1)? By the way, I cannot see an increase in precipitation after 120 ka! Please clarify.

Chapter 5.3. It would be nice to see a direct comparison of climate parameters between Lake Ohrid, LGdM, Ioannina, and the records in France (e.g., in Figure 5). Unfortunately, the southern European records are only summarized in Figure 6. I think it would be helpful for your argumentation. Be careful with simplification of complex interactions! When I am looking at the comparison between LO, Northern Europe, and Southern Europe (Figure 6), I can recognize several different responses to global climate changes in all records. In my opinion, the authors should make it unequivocally clear the transitional position from Mediterranean climate influenced climate to more temperate northern European climate conditions with, e.g., a distinct temperature decrease after 125 ka, which is not that pronounced at LO (more comparable to the southern European records).

P11, 26. The period from 135-105 ka comprises the late MIS 6 to MIS 5c, as you already mentioned it in the next sentence!

P11, 29. In the direct comparison between LO and Grande Pile & Bouchet, there are opposite trends in the anomalies at the end of MIS 6. Between ca. 140-133 ka: high anomalies at LO, low at GP; between ca. 133-128 ka: low anomalies at LO, high at GP. Please clarify.

P12, 11. Delete the repetition of 'Fig.5'.

P12, 18. As I already wrote above, add the 'other pollen records from Lake Ohrid' to the figures. It would be helpful for the following of your argumentation.

P12, 24-35. What is the 'striking feature' of these interstadials, just the occurrence? Add some more explanations. I think these two short-term interstadial can be correlated with the Dansgaard-Oeschger events DO 19 and 20, which are also visible in the eastern Mediterranean records, such as Thenaghi Philippon (Müller et al., 2011) and Lake Van (Pickarski et al., 2015), even though the climate was significantly more continental during this time.

P13, 17 & 23. The ODP-976 record is not presented in the manuscript!

P13, 16-24. There are some important differences visible between LO and other records (e.g., at ODP-977, Villars cave) especially at the early Eemian, which are not discussed in the text. Be careful with generalization! Please add more details and discussed that differences a bit more.

P13, 25 -34. There are some logical steps missing. Which event centered at ca. 115? Melisey I? C25? I am a bit lost in this section! In addition, C25 event is not visible in

the SST record! Please clarify!

P14, 5. A period is missing at the end of the sentence.

P14, 8. Insert '...Last Interglacial Complex (LIC, 128 to 70 ka),...' due to the used abbreviation in the next sentence.

P14, 12. '...occurring during the late MIS 6, MIS 5 and the early MIS 4.'

Table 1 Please use a uniform nomenclature. It would be nice if you could mark the different MIS 5 stages (MIS 5e to a) in the 'Marine Statigraphy' column.

Figures

Figure 1 Where is the 'Struga meterological station' located? Can you mark it on the map? Please pay more attention to the consistence of facts between the text and the figures. For example, you mentioned in P4, 7 that Lake Ohrid is located at 693 m asl. In your figure 1, it is written 694 m asl. The same discrepancy is evident in the mean annual temperature at Lake Ohrid (P4, line 15).

Figure 2 Perhaps it is better to use the terms 'Mesophilous taxa/biome' and 'Mediterranean taxa/biome' instead of 'trees', because Hedera is not a tree, it is a liane, and Cistus (depending on the species) grows also as shrub. The figure caption is a bit confusing. If you are showing, e.g., only Poaceae within the group of grasslands than delete the additional information that grasslands consist of Poaceae and Cyperaceae. The same goes for 'Steppe'. Please, show in the first column (left) the MIS 6 to 4 and in the second column the nomenclature of the northern and central Europe. That goes also for the other figures.

Figure 3 MAT method is shown in a blue line, not in black! Regarding GDD5: The legend of the figure is not clear to me. Are these 1000-3000 years over 5°C per year/season/? Are these 1000-3000°C. Please, clarify! Delete the repetition of 'Blue shading indicates cold periods (Riss glacial and Early Würm glacial stadials)' in the figure caption.

Figure A1 What do you mean with 'The last graph represents the ....'? Figure A2? What outline the different red lines? More details are needed.

References

Müller et al., 2011. The role of climate in the spread of modern humans into Europe, QSR 30. 273-279.

Pickarski et al., 2015. Abrupt climate and vegetation variability of eastern Anatolia during the last glacial. CP 11. 1491-1505.

Rasmussen et al., 2014. A stratigraphic framework for abrupt climatic changes during the Last Glacial period based on three synchronized Greenland ice-core records: refining and extending the INTIMATE event stratigraphy. QSR 106. 14-28.

Tzedakis, 2007. Seven ambiguities in the Mediterranean palaeoenvironmental narrative. QSR 26. 2042-2066.

I hope my comments help improving the manuscript.

―――――――――――――――――――――――――――――

---

## Referee Comment (RC3) · Anonymous Referee #3 · 28 Oct 2018

The quantitative palaeoclimate reconstructions during the Quaternary is an important to understand the climate changes and its potential forcing mechanisms, thus can helpful for predicting climate changes in future global warming. In this study, a quantitative reconstruction of climate parameters was provided based on the pollen data from the Lake Ohrid in southern Europe, using two complementary approaches including 'Modern Analogues Technique' and 'Weighted Averaging Partial Least-Squares Regressio' during the period from 160 to 70 ka. It is useful for better understanding climatic changes during the key periods of MIS 6 and MIS 5 in the South Europe.

In current version I would suggest a minor revision before accepting it for publication.

[Figure]

Here are a few basic comments that could guide the authors to submit a more detailed manuscript.

1. For the reconstruction, Pinus has been excluded in this study due to its overwhelming presence in the DEEP would potentially masks climatically controlled environmental signals from other taxa. Because this change should effect on the quantitative reconstruction, so a more detail comparison on reconstruction of climate parameters between Pinus including and its excluding is best presented in the supplementary information. 2. In Figure A2, the most values of Squared-chord distance between the first and the last analogue for a chosen climate parameter (TANN) calculated by MAT method are more than 0.3. The values may be larger than the no-analog/analog threshold that could accurate and precise palaeoclimate reconstruction. Therefore, a systematically analysis between the Squared-chord distance and precision of palaeoclimate reconstruction need to employ. 3. Results suggest that the Lake Ohrid palaeoclimate reconstruction shows greater similarity with climate patterns inferred from northern European pollen records than with southern European records in figure 6. Because the Lake Ohrid locates in the southern European, thus a more detail explanation and possible mechanism should be mentioned. 4. Please check all the language in the text, and correct it to the English, e.g. page 5 lines 1-3, and page 6 lines 35-36.

---

## Author Comment (AC1) · 5 Nov 2018

Referee: This is a very interesting paper worthy to be published and here are my suggestions for some improvements following the continuity of the text.

P1, Line 25 (and p.14 lines 13-17). For me, the discussion on the stability of the Eemien rests on a misdefinition. Since Jessen, everybody know that an interglacial cycle includes a period of warming after the previous glacial, an optimum and finally a progressive cooling leading to the next glacial. In this way the instability you mention is a truism! You could just mention that "The Eemian in the Balkans was characterized by an abrupt early warming during its anathermic phase followed a central phase .

. ..). Most of the authors, when dealing with "instability" or "stability" try to identify some short terms climate "oscillations" disturbing the classical interglacial trend on a warming followed by a cooling (as well discussed latter in your paper).

AUTHORS: THE REFEREE IS RIGHT SAYING THAT THE "INSTABILITY" OF THE EEMIAN IS A WELL ASSESSED SUBJECT. WE WILL TRY TO MAKE CLEARER AND CORRECT IT.

Referee: P2, l.30-35. Before Field et al. (1994), at least Beaulieu & Reille (1984,1889) already mentioned a period of transitional warming during the late Eemian.

A: WE WILL CONSIDER THIS

Referee: P3, l.2 and fig. 6: be careful with the coordinates. Les Echets: 45âŮę52'36" N, 4âŮę55'44" E and Lac du Bouchet : 44âŮę54'31" N , 3âŮę47'30". Lac du Bouchet is really transitional between North and South according to your classification. As a matter of fact, this limit at 45âŮęN is of interest as far as the Eemian is concerned, but during the Early Würm the story is more complex with the opposition between the "Odderade style" vegetation and climate successions and the "Grande Pile style" successions. It could be mentioned.

A: THANK YOU, WE WILL CONSIDER THIS

Referee: P3, l.27. : I should prefer "higher resolution", as an interval of 400 is not high resolution in terms of vegetation dynamics. Would you accept a pollen diagram covering the whole Holocene in only 28 spectra?

A: WE AGREE, IT'S "HIGHER RESOLUTION"

Referee: P4, the first sentence is not necessary as the following description is sufficient.

A: OK

Referee: P4, l.11: do not repeat "karst aquifers".

A: OK

Referee: P5, top: my copy is polluted by three lines in pseudo-latin.

A: OK, we will delete the odd phrase

Referee: P5, l.34: could you develop and explain in more details your choice of 6 modern analogues?

A: OK

Referee: P6, l. 20: I suggest "pollen records " instead of "data" P6, l.35: again one line polluted with latin.

A: WE PROPOSE TO CHANGE INTO LOW-RESOLUTION DATA, THE POLLEN RECORD IS THE SAME

Referee: P9, l.5 and after: this interesting discussion could be included in the chapter "Materials and methods"?

A: YES, WE WILL ANTICIPATE THIS IN THE MATERIALS AND METHODS

Referee: P9, l.22: may-be a clumsy statement. If your climate reconstructions are derived from pollen spectra, it would be a great disaster for your results if they were not in agreement

A: OK, L.26 - WE WILL DELETE POLLEN

Referee: P10, l. 24-29: Very interesting but I do not understand how the discussion is inferred from fig. 4: TIC and TOC do not change (slight increase in TOC) during the interval between 137 and 135 Ka marked by high amplitude changes in PAN and TAN??

A: WE ARE USING DATA FROM OTHER CORES AND OTHER PAPERS. WE WILL TRY TO MAKE IT MORE CLEAR

Referee: P10, l 34 : Not that slight??

A: SEE COMMENTS OF REFEREE 2, QUITE THE OPPOSITE

Referee: Chapter 5.4. : it would be of interest to take into account the climate reconstructions based on diatoms populations established by Rioual et al. (2007) at Ribains ( see also Shemesh et al., 2001)

A: OK, THANK YOU FOR THE SUGGESTIONS. WE KNOW THE PAPER AND ALREADY USED IT IN OUR INTERPRETATION

References Beaulieu J.-L. de & Reille M., 1984. A long upper Pleistocene pollen record from Les Echets near Lyon, France, Boreas, 13, p.111-132. Beaulieu J.-L. de & Reille M., 1989. The transition from temperate phases to stadials in the long Upper Pleistocene sequence from les Echets (France). Palaeogeography, ,Palaeoclimatology, Palaeoecology, 72, 147-159. Rioual P.., Andrieu-Ponel V., Beaulieu J.-L. de, Reille M., Svobodova H. & Battarbee R. W., 2007. Diatom responses to limnological and climatic changes at Ribains maar (French Massif Central) during the Eemian and Early Würm. Quaternary Science Re- views, 26 (11-12), 1557-1609. Shemesh A., Rietti-Shati M., Rioual P., Battarbee R., Beaulieu J.-L. de, Reille M. and Svobodova H., 2001. An Oxygen isotope record of lacustrine opal from a european Maar indicates climatic stability during the last interglacial. Geophysical Research Let- ters, 28 (12), 2305-2308.

---

## Author Comment (AC2) · 5 Nov 2018

Referee: Thank you for the opportunity to comment on this manuscript. The manuscript titled 'Pollen-based temperature and precipitation changes in the Ohrid Basin (western Balkans) between 160 and 70 ka' covers a sound data set of great value to the palaeoecological community as climate reconstruction for the last interglacial-glacial cycle from this sensitive area are scarce. Overall, I think the work is good and should be published in CP but there are a number of important details that need to be considered and corrected first. In many instances these are related to terminology, definition of terms and ambiguity or circularity in the phrasing. One important example of this is the use

of different nomenclatures, e.g., from alpine region (Riss/Würm glacial), from north-ern/central Europe (Eemian interglacial), and the special nomenclature from France for interstadials/stadials. Regarding the last interglacial, be careful with the statement that the Eemian was not a stable phase in the Balkan region. The last interglacial at LO clearly shows a classical interglacial with an early warming at the beginning of MIS 5e, a climate optimum and a progressive cooling towards the end of the last interglacial. In general, I suggest to interpret the LO record with regard to further regional climate reconstructions (mentioned in the chapter 'Introduction', e.g., LGdM and Ioannina) and use it as a basis for discussing in more global scale with possible correlations to the France, speleothem records, MIS, etc. These important issues and more are detailed below along with some suggestions for grammatical corrections.

Page, Line. Comment P1, 17. The presented archive covers the period between 160 to 70ka. This includes not only MIS 6 and MIS 5, but also the early part of MIS 4. The authors mentioned it in the conclusion by themselves. (P14, 8).

AUTHORS: OK

Referee: P1, 25-28. According to the anonymous referee #1. The last interglacial at Lake Ohrid shows a classical interglacial cycle with pre-temperate phase (early warm-ing), temper- ate phase (climate optimum), and a post-temperate phase (progressive cooling), which is also well described by Tzedakis, 2007. Be careful with this general conclusion of an unstable last interglacial in the Balkan region!

A: OK, WE WILL ADJUST THE TEXT

Referee: P2, 3. Insert '...'MIS' – 6 (penultimate glacial) and MIS 5 (last interglacial complex) are. . ..'

A: OK

Referee: P2, 6. '. . .the penultimate glacial (or Riss Glaciation) . . .' in comparison to P2, 12. 'The Eemian. . .' Please pay more attention to a uniform nomenclature.

The term 'Riss glaciation' is normally used in the alpine area. In northern and central Europe, the penultimate glacial belongs to the 'Late Saale/Saalian Complex'. If you want to continue with the term 'Eemian' for the last interglacial (MIS 5e), you should use the nomenclature of the northern and central Europe. Another example is the '(Early) Weichselian glacial' instead of 'Early Würm' (P2, 34).

A: OK

Referee: P2, 7. Please pay more attention to uniformity. The LIC lasts from ca. 130-80 ka in the chapter 'Introduction', whereas the LIC covers the period from ca. 128-70 ka at page 3, 25. Please check the dates, there are several more discrepancies.

A: OK, WE MADE IT CLEARER. THE MIS 6 TO MIS 5 TRANSITION AT LAKE OHRID HAS BEEN THE SUBJECT OF AN ACCURATE ALIGNMENT AND SYNCHRONIZA-TION (ZANCHETTA ET AL., 2016) RESULTING IN AN OFFSET OF 2 KA WITH OTHER RECORDS (E.G. GOVIN ET AL., 2015; RAILSBACK ET AL., 2015).

Referee: P3, 2-4. The authors mentioned that Ioannina and LGdM have done climate reconstructions based on pollen data. Unfortunately, these two archives were not used for comparison (e.g. in Figure 5) or were not discussed in detail in the text (chapter 5.3), although these records are much closer to Lake Ohrid than the archives in France. Due to the fact that you mentioned in P3, line 15 that the Balkan Peninsula is a key region between the Mediterranean area and the Northern/Central Europe. It would be nice to see how these few southern European records differ from the northern European ones. What about the direct comparison with Lake Prespa, which covers the last âĹij90 ka. I am not sure if they have done climate reconstructions, but are there any similarities or differences to your record?

A: THERE IS NOT A CLIMATE RECONSTRUCTION FOR LAKE PRESPA, AND THE CHRONOLOGY FOR BOTTOM CORE WILL BE PROBABLY IMPROVED IN THE FU-TURE. IOANNINA AND LGDM DO NOT HAVE SINGLE CLIMATE RECONSTRUC-TIONS. THE POLLEN DATA OF THE TWO RECORDS JUST CONTRIBUTED TO THE

SYNTHETIC CURVES SHOWN IN BREWER ET AL. 2008.

Referee: P3, 14. . . .glacial-interglacial cycle. (?)

A: OK

Referee: P4, 1-2. The first sentence is not necessary.

A: OK

Referee: P4, 11. Avoid the repetition of 'karst aquifers' at the end of line 11.

A: OK

Referee: P4, 12-13. Rephrase: '. . .small streams, rivers, and by direct precipitation.'

A: OK

Referee: P4, 18. Rephrase: '...during winter and south-southeasterly (or southerly to south- easterly) winds during. . ..'

A: OK

Referee: P4, 21. This context is not clear – please rephrase. What are the four zones? Which species dominate which zone?

A: OK, WE WILL CORRECT IT

Referee: P5, 1-3. Please check. This sentence is written in a different language.

A: OK, WE WILL DELETE IT Referee: P5, 15. What is 'ne w' in the high-resolution pollen data, presented in this manuscript, when it is already published in Sinopoli et al., 2018? Did you analyse more 'new' samples for this manuscript, which are not shown in the Sinopoli et al. paper? Please clarify!

A: OK, WE WILL MAKE IT CLEARER.

Referee: P5, 34. Which 'six modern analogues'? This subject should be further explained and clarified in the text.

A: OK

Referee: P6, 18. It is not clear to me what do you mean with the '. . .first analogue and the last analogue. . .'? More details are needed.

A: OK

Referee: P6, 35-36. Please check. This sentence is written in a different language.

A: OK, WE WILL DELETE IT

Referee: P7, 1. '. . .and annual precipitation between 350 and 600 mm/yr),. . .' It depends on what method you looking at. For MAT, I can recognize a fluctuation from 100 to ca. 300 mm/yr in the mean annual precipitation. For WALPS, it fluctuates between 500 to 700 mm/yr. What is the explanation for this huge difference? Please clarify and add some more explanations.

A: OK, WE TRIED TO EXPLAIN THIS, AND WE WILL IMPROVE OUR EXPLANATION.

Referee: P7, 11 and 14. Which 'other methods'? If necessary, add references.

A: WE ARE NOT SURE WE SHOULD LIST THEM, AS WE REFER TO THE ORIGINAL PUBLICATIONS

Referee: P7, 34-36. Which is the third part? Furthermore, an additional verb is missing. Please rephrase this sentence.

A: MELISEY II AS SHOWN IN TABLE 1, YES "ARE" IS MISSING. WE WILL ADJUST IT

Referee: P8, 17 and following. Describe the 'end of MIS 6' within chapter 4.1

A: YES, HERE IS JUST USED AS A COMPARISON

Referee: P9, 8 -16. This section should be mentioned in the chapter 'Materials &

methods'.

A: OK, WE WILL MOVE IT

Referee: P9, 19-20. To be consistent with the text, could you add the discussed pioneer shrubs (e.g. Juniperus) to the selected pollen diagram (Figure 2).

A: OK

Referee: P9, 28 and following. For the better understanding and demonstration, it would be very helpful to show the comparison of your Eemian climate reconstructions with those from the JO2004 record. Please insert the JO2004 climate reconstruction, for example, in Figure 4.

A: THERE IS NOT A CLIMATE RECONSTRUCTION IN THE JO2004. THE RECORD WAS USED FOR A CUMULATIVE RECONSTRUCTION BY BREWER ET AL. (2008).

Referee: P10, 12. The phrase '. . .export of terrestrial organic matter. . .' sounds a bit odd. Please rephrase.

A: OK

Referee: P10, 18 and 19. Delete '. . .inferred from pollen.' due to the repetition from the previous sentence. At this point, it is obvious that TANN and PANN were calculated from pollen.

A: OK

Referee: P10, 18-29. There seem to be some logical steps missing. I cannot work out how lake level changes can be visible in the pollen record. I also cannot see a decline in terrestrial vegetation at the end of MIS 6 - in fact, quite the opposite. It shows a continuous increase in mesophilous and coniferous trees! In addition, I assume that significant lake level changes should be reflected in the TIC /TOC values, but again I cannot see any changes in these proxies at the DEEP site. Furthermore, the 'clearly seen' change in the pollen record of Co2012 is not presented in this manuscript! These

subjects should be explained more clearly in the text.

A: THE REFEREE IS RIGHT, LAKE LEVEL CHANGES ARE NOT VISIBLE IN THE POLLEN RECORD! WE USED OTHER EVIDENCES FROM PUBLISHED ARTICLES: "THE DISTINCT HIGH-AMPLITUDE FLUCTUATIONS INFERRED FROM POLLEN DURING THE FINAL PART OF MIS 6, COULD AT LEAST PARTLY BE DUE TO LAKE LEVEL CHANGES AS THE WATER TABLE DURING THIS PERIOD WAS GENER-ALLY ON THE RISE (LINDHORST ET AL., 2010, HOLTVOETH ET AL., 2017; WAG-NER ET AL., 2017)". THIS IS IN VERY GOOD AGREEMENT WITH "A CONTINUOUS INCREASE IN MESOPHILOUS AND CONIFEROUS TREES" AS EVIDENCED BY REFEREE 2. ANYWAY IT APPEARS THAT WE HAVE NOT BEEN CLEAR ENOUGH, WE WILL TRY TO BE MORE EFFECTIVE.

Referee: P10, 31-34. In my opinion, a difference of 500 years is NOT a discrepancy. The authors should soften the language.

A: WE THOUGHT THAT USING "SLIGHT" COULD HAVE BEEN ENOUGH, AND IT IS NOT FOR REFEREE 2. . . BUT REFEREE 1 SAYS IS NOT THAT SLIGHT. . . SO THE OPPOSITE "THIS SLIGHT DISCREPANCY" (NAMELY 500 YEARS) "IS PROBABLY DUE TO DIFFERENCES IN THE CHRONOLOGY ESTABLISHED FOR THE TWO CORES". WE WILL EMPHASIZE THAT THE TWO CHRONOLOGIES HAVE BEEN ASSESSED INDEPENDENTLY.

Referee: P10, 39-41. Please avoid the use of so many 'and' in this sentence. Please rephrase.

A: OK

Referee: P10, 42 and following. There is something odd about the line of reasoning here. It is not clear to me what do the authors mean with 'from 120 ka and culminating at 119.4 ka'? In Fig. 4, I cannot identify a 'culmination' in the TIC decrease during this interval. At the DEEP site, the TIC and TOC values already continuously decrease

after ca. 126 ka! In addition, how can a progressive drying (P11, 3) take place when precipitation increases at the same time (P11, 1)? By the way, I cannot see an increase in precipitation after 120 ka! Please clarify.

A: OK

Referee: Chapter 5.3. It would be nice to see a direct comparison of climate parameters between Lake Ohrid, LGdM, Ioannina, and the records in France (e.g., in Figure 5). Unfortunately, the southern European records are only summarized in Figure 6. I think it would be helpful for your argumentation. Be careful with simplification of complex interactions! When I am looking at the comparison between LO, Northern Europe, and Southern Europe (Figure 6), I can recognize several different responses to global climate changes in all records. In my opinion, the authors should make it unequivocally clear the transitional position from Mediterranean climate influenced climate to more temperate northern European climate conditions with, e.g., a distinct temperature decrease after 125 ka, which is not that pronounced at LO (more comparable to the southern European records).

A: FOR IOANNINA NO SINGLE CURVES FOR CLIMATE RECONSTRUCTION ARE AVAILABLE, THE ORIGINAL DATA HAVE BEEN USED TO BE INCLUDED IN SUMMARY CURVES (BREWER ET AL., 2008). FOR LGDM THE ELABORATION OF DATA PUBLISHED BY ALLEN ET AL. (2000) IS QUITE DIFFERENT AND NOT DIRECTLY COMPARABLE. THIS IS WHY WE HAVE BEEN ABLE TO USE ONLY THE SUMMARIZED POLLEN-BASED RECONSTRUCTION OF BREWER ET AL. IN FIG. 6

Referee: P11, 26. The period from 135-105 ka comprises the late MIS 6 to MIS 5c, as you already mentioned it in the next sentence!

A: YES, THE REFEREE IS RIGHT. WE USED IN FACT THE VERB "INCLUDES", MEANING THAT IS NOT ONLY THE WHOLE EEMIAN: "THE PERIOD 135- 105 KA, WHICH INCLUDES THE WHOLE EEMIAN"

Referee: P11, 29. In the direct comparison between LO and Grande Pile & Bouchet, there are opposite trends in the anomalies at the end of MIS 6. Between ca. 140-133 ka: high anomalies at LO, low at GP; between ca. 133-128 ka: low anomalies at LO, high at GP. Please clarify.

A: OK

Referee: P12, 11. Delete the repetition of 'Fig.5'.

A: OK

Referee: P12, 18. As I already wrote above, add the 'other pollen records from Lake Ohrid' to the figures. It would be helpful for the following of your argumentation.

A: THE REFEREE IS RIGHT, BUT THE DATA ARE NOT AVAILABLE TO US

Referee: P12, 24-35. What is the 'striking feature' of these interstadials, just the occurrence? Add some more explanations. I think these two short-term interstadial can be correlated with the Dansgaard-Oeschger events DO 19 and 20, which are also visible in the eastern Mediterranean records, such as Thenaghi Philippon (Müller et al., 2011) and Lake Van (Pickarski et al., 2015), even though the climate was significantly more continental during this time.

A: YES, THEY CAN BE PROBABLY CORRELATED WITH THE TE DANSGAARD-OESCHGER EVENTS DO 19 AND 20. WE DO NOT UNDERSTAND WITH" EVEN THOUGH THE CLIMATE WAS SIGNIFICANTLY MORE CONTINENTAL DURING THIS TIME"

Referee: P13, 17 & 23. The ODP-976 record is not presented in the manuscript!

A: YES, THE REFEREE IS RIGHT, IT'S A MISTAKE

Referee: P13, 16-24. There are some important differences visible between LO and other records (e.g., at ODP-977, Villars cave) especially at the early Eemian, which are not discussed in the text. Be careful with generalization! Please add more details and

discussed that differences a bit more.

A: YES, THERE ARE DIFFERENCES DIFFICULT TO EXPLAIN. WE DON'T ANYWAY THINK IT IS THE CASE TO GO INTO SUCH DETAILS.

Referee: P13, 25 -34. There are some logical steps missing. Which event centered at ca. 115? Melisey I? C25? I am a bit lost in this section! In addition, C25 event is not visible in the SST record! Please clarify!
 A: OK, WE WILL CLARIFY THIS PART.

Referee: P14, 5. A period is missing at the end of the sentence.

A: OK, WE WILL CLARIFY THIS PART.

Referee: P14, 8. Insert '...Last Interglacial Complex (LIC, 128 to 70 ka),...' due to the used abbreviation in the next sentence.

A: OK. JUST IN CASE SOME READERS ARE JUST READING THE CONCLUSIONS.

Referee: P14, 12. '. . .occurring during the late MIS 6, MIS 5 and the early MIS 4.'
Table 1 Please use a uniform nomenclature. It would be nice if you could mark the different MIS 5 stages (MIS 5e to a) in the 'Marine Statigraphy' column.

A: WE WILL DO A ROUGH SCHEME, BUT AS FAR AS WE KNOW THE DIRECT CORRESPONDENCE BETWEEN TERRESTRIAL PHASES AND MARINE STRATIG-RAPHY WAS NOT YET PRECISELY ESTABLISHED.

Figures Referee: Figure 1 Where is the 'Struga meterological station' located? Can you mark it on the map? Please pay more attention to the consistence of facts between the text and the figures. For example, you mentioned in P4, 7 that Lake Ohrid is located at 693 m asl. In your figure 1, it is written 694 m asl. The same discrepancy is evident in the mean annual temperature at Lake Ohrid (P4, line 15).

A: OK, THANK YOU

Referee: Figure 2 Perhaps it is better to use the terms 'Mesophilous taxa/biome' and

'Mediterranean taxa/biome' instead of 'trees', because Hedera is not a tree, it is a liane, and Cistus (depending on the species) grows also as shrub. The figure caption is a bit confusing. If you are showing, e.g., only Poaceae within the group of grasslands than delete the additional information that grasslands consist of Poaceae and Cyperaceae. The same goes for 'Steppe'. Please, show in the first column (left) the MIS 6 to 4 and in the second column the nomenclature of the northern and central Europe. That goes also for the other figures.

A: WE USED THE SAME CATEGORIES AND NAMES USED IN SINOPOLI ET AL. 2018 AND SADORI ET AL. 2016.

Referee: Figure 3 MAT method is shown in a blue line, not in black! Regarding GDD5: The legend of the figure is not clear to me. Are these 1000-3000 years over 5ÂŮęC per year/season/? Are these 1000-3000◦C. Please, clarify! Delete the repetition of 'Blue shading indicates cold periods (Riss glacial and Early Würm glacial stadials)' in the figure caption.

A: OK, WE WILL ADJUST IT

Referee: Figure A1 What do you mean with 'The last graph represents the . . ..'? Figure A2? What outline the different red lines? More details are needed.

A: OK, WE WILL PRECISE THIS

References Müller et al., 2011. The role of climate in the spread of modern humans into Europe, QSR 30. 273-279. Pickarski et al., 2015. Abrupt climate and vegetation variability of eastern Anatolia during the last glacial. CP 11. 1491-1505. Rasmussen et al., 2014. A stratigraphic framework for abrupt climatic changes dur- ing the Last Glacial period based on three synchronized Greenland ice-core records: refining and extending the INTIMATE event stratigraphy. QSR 106. 14-28. Tzedakis, 2007. Seven ambiguities in the Mediterranean palaeoenvironmental narra- tive. QSR 26. 2042-2066. I hope my comments help improving the manuscript.

YES, THEY DID. THANK YOU VERY MUCH

---

## Author Comment (AC4) · 5 Nov 2018

The quantitative palaeoclimate reconstructions during the Quaternary is an important to understand the climate changes and its potential forcing mechanisms, thus can helpful for predicting climate changes in future global warming. In this study, a quantitative reconstruction of climate parameters was provided based on the pollen data from the Lake Ohrid in southern Europe, using two complementary approaches including 'Mod- ern Analogues Technique' and 'Weighted Averaging Partial Least-Squares Regressio' during the period from 160 to 70 ka. It is useful for better understanding climatic changes during the key periods of MIS 6 and MIS 5 in the South Europe. In

current version I would suggest a minor revision before accepting it for publication. C1 Here are a few basic comments that could guide the authors to submit a more detailed manuscript.

Referee: 1. For the reconstruction, Pinus has been excluded in this study due to its overwhelming presence in the DEEP would potentially masks climatically controlled environmental signals from other taxa. Because this change should effect on the quantitative reconstruction, so a more detail comparison on reconstruction of climate parameters between Pinus including and its excluding is best presented in the supplementary information.

AUTHORS: OK, WE WILL TAKE THE ADVICE INTO CONSIDERATION

Referee: 2. In Figure A2, the most values of Squared-chord distance between the first and the last analogue for a chosen climate parameter (TANN) calculated by MAT method are more than 0.3. The values may be larger than the no-analog/analog threshold that could accurate and precise palaeoclimate reconstruction. Therefore, a systematically analysis between the Squared-chord distance and precision of palaeoclimate reconstruction need to employ.

A: OK, WE WILL TAKE THE ADVICE INTO CONSIDERATION

Referee: 3. Results suggest that the Lake Ohrid palaeoclimate reconstruction shows greater similarity with climate patterns inferred from northern European pollen records than with southern European records in figure 6. Because the Lake Ohrid locates in the southern European, thus a more detail explanation and possible mechanism should be mentioned.

A: OK, WE WILL TRY TO CLARIFY OUR SUPPOSITION

Referee: 4. Please check all the language in the text, and correct it to the English, e.g. page 5 lines 1-3, and page 6 lines 35-36.

A: SURE, THANK YOU

---

## Author Comment (AC5) · 7 Nov 2018

After a debate among authors, we decide to reply more in detail to the referees clarifying how we intend to improve the manuscript. We would ask the editor and the referees to consider this version.

This is a very interesting paper worthy to be published and here are my suggestions for some improvements following the continuity of the text. Referee: P1, Line 25 (and p.14 lines 13-17). For me, the discussion on the stability of the Eemien rests on a misdefinition. Since Jessen, everybody know that an interglacial cycle includes a period of warming after the previous glacial, an optimum and finally a progressive cooling

leading to the next glacial. In this way the instability you mention is a truism! You could just mention that "The Eemian in the Balkans was characterized by an abrupt early warming during its anathermic phase followed a central phase . . ..). Most of the authors, when dealing with "instability" or "stability" try to identify some short terms climate "oscillations" disturbing the classical interglacial trend on a warming followed by a cooling (as well discussed latter in your paper).

Authors: THE REFEREE IS RIGHT SAYING THAT THE "INSTABILITY" OF THE EEMIAN IS A WELL ASSESSED SUBJECT. WE WILL TRY TO MAKE CLEARER AND CORRECT IT.

P2, l.30-35. Before Field et al. (1994), at least Beaulieu & Reille (1984,1889) already mentioned a period of transitional warming during the late Eemian.

WE WILL CONSIDER THIS

P3, l.2 and fig. 6: be careful with the coordinates. Les Echets: 45°52'36"N, 4°55'44"E and Lac du Bouchet: 44°54'31"N , 3°47'30" Lac du Bouchet is really transitional between North and South according to your classification. As a matter of fact, this limit at 45°N is of interest as far as the Eemian is concerned, but during the Early Würm the story is more complex with the opposition between the "Odderade style" vegetation and climate successions and the "Grande Pile style" successions. It could be mentioned.

THANK YOU, WE WILL CONSIDER THIS

P3, l.27. : I should prefer "higher resolution", as an interval of 400 is not high resolution in terms of vegetation dynamics. Would you accept a pollen diagram covering the whole Holocene in only 28 spectra?

WE AGREE, IT'S "HIGHER RESOLUTION"

P4, the first sentence is not necessary as the following description is sufficient.

OK

P4, l.11: do not repeat "karst aquifers".

OK

P5, top: my copy is polluted by three lines in pseudo-latin.

OK, WE WILL DELETE THE ODD PHRASE

P5, l.34: could you develop and explain in more details your choice of 6 modern analogues?

TEXT WILL BE CHANGED "THE NUMBER OF ANALOGUES USED MAY AFFECT THE QUALITY OF THE RECONSTRUCTIONS. IN THE PRESENT PAPER, THE MOST ROBUST RECONSTRUCTIONS ARE OBTAINED USING SIX ANALOGUES. IT'S THE OPTIMAL NUMBER OF ANALOGUES (DETERMINED USING THE LOWEST ROOT-MEAN-SQUARE ERROR OF PREDICTION) TO MINIMIZE THE CHANCES OF FALSELY DETERMINING TWO MODERN SAMPLES TO BE ANALOGUES OR CONSIDERING TWO ANALOGOUS SAMPLES NOT TO BE ANALOGUES"

P6, l. 20:I suggest "pollen records " instead of "data" P6, l.35: again one line polluted with latin.

WE PROPOSE TO CHANGE INTO LOW-RESOLUTION DATA, THE POLLEN RECORD IS THE SAME

P9, l.5 and after: this interesting discussion could be included in the chapter "Materials and methods"?

WE DON'T AGREE: THIS SECTION TRY TO EXPLAIN THE DIFFERENCES IN THE RESULTS DUE TO THE DIFFERENT METHODS USED; SO IT'S REALLY BETTER TO KEEP IT IN THE DISCUSSION PART

P9, l.22: may-be a clumsy statement. If your climate reconstructions are derived from pollen spectra, it would be a great disaster for your results if they were not in agreement

OK, L.26 - WE WILL DELETE POLLEN

P10, l. 24-29: Very interesting but I do not understand how the discussion is inferred from fig. 4: TIC and TOC do not change (slight increase in TOC) during the interval between 137 and 135 Ka marked by high amplitude changes in PAN and TAN??

WE ARE USING DATA FROM OTHER CORES AND OTHER PAPERS. GENER-ALLY, LAKE-LEVEL CHANGE RESPONDS TO PRECIPITATION CHANGE RATHER THAN TEMPERATURE AND MAY THEREFORE HAVE AFFECTED THE DISTRIBU-TION OF LOW-LYING TERRESTRIAL HABITATS ON FLAT TERRACE SURFACES BEFORE THE TEMPERATURE THRESHOLD WAS CROSSED THAT LED TO IN-CREASING IN-LAKE TIC PRECIPITATION. TOC FROM COMBINED AQUATIC AND TERRESTRIAL SOURCES IS A MINOR FRACTION OF THE TOTAL SEDIMENT AND, AS SUCH, IS SUSCEPTIBLE TO DILUTION BY LITHOGENIC MATERIAL. THUS, IF PRECIPITATION INCREASED THE SUPPLY OF BOTH TERRESTRIAL OC AND LITHOGENIC (SILICICLASTIC) MATERIAL WOULD HAVE INCREASED. DEPEND-ING ON THE MAIN OC SOURCES THIS CAN LEAD TO BOTH AN INCREASE OR DECREASE IN TOC WITH INCREASING EROSION RATES, OR TOC MAY IN FACT REMAIN TEMPORARILY CONSTANT IF ITS MAIN SOURCE IS, E.G., SOIL OC (INCREASE OF OC AND SILICICLASTIC MATERIAL AT SAME RATE). THUS, WE DON'T NECESSARILY EXPECT TO SEE A COVARIATION OF PANN AND TANN WITH TIC AND TOC AT THIS TRANSITIONAL STAGE. WE WILL TRY TO MAKE IT MORE CLEAR

P10, l 34 : Not that slight??

SEE COMMENTS OF REFEREE 2, QUITE THE OPPOSITE

Chapter 5.4. : it would be of interest to take into account the climate reconstructions based on diatoms populations established by Rioual et al. (2007) at Ribains (see also Shemesh et al., 2001)

[Figure]

THANKS FOR THE SUGGESTION. HOWEVER IT'S NOT POSSIBLE TO ADD IT IN THE FIGURES (E.G. FIG. 5) BECAUSE IN THE RIOUAL ET AL (2007) PAPER, THE TEMPERATURE RECONSTRUCTION BASED ON THE DIATOMS IS NOT DIRECTLY COMPARABLE TO THE OHRID ONES. RIOUAL ET AL (2007) PLOTTED PDCA SAMPLE SCORES POLLEN DIATOMS AS A PROXY OF TEMPERATURE, NOT TEMPERATURE VALUES. IN THE PAPER OF RIOUAL ET AL (2001), THE CLIMATE RECONSTRUCTION BASED ON RIBAINS POLLEN DATA CANNOT BE ADDED HERE IN OUR FIGURE AS ITS PLOTTED AS FUNCTION OF DEPTH, NOT OF AGE. BUT, OF COURSE, THESE TWO PAPERS ARE CITED AND USED IT IN OUR INTERPRETATION

References Beaulieu J.-L. de & Reille M., 1984. A long upper Pleistocene pollen record from Les Echets near Lyon, France, Boreas, 13, p.111-132. Beaulieu J.-L. de & Reille M., 1989. The transition from temperate phases to stadials in the long Upper Pleistocene sequence from les Echets (France). Palaeogeography, ,Palaeoclimatology, Palaeoecology, 72, 147-159. Rioual P.., Andrieu-Ponel V., Beaulieu J.-L. de, Reille M., Svobodova H. & Battarbee R. W., 2007. Diatom responses to limnological and climatic changes at Ribains maar (French Massif Central) during the Eemian and Early Würm. Quaternary Science Re- views, 26 (11-12), 1557-1609. Shemesh A., Rietti-Shati M., Rioual P., Battarbee R., Beaulieu J.-L. de, Reille M. and Svobodova H., 2001. An Oxygen isotope record of lacustrine opal from a european Maar indicates climatic stability during the last interglacial. Geophysical Research Let- ters, 28 (12), 2305-2308.

---

## Author Comment (AC6) · 7 Nov 2018

After a debate among authors, we decide to reply more in detail to the referees clarifying how we intend to improve the manuscript. We would ask the editor and the referees to consider this version.

Thank you for the opportunity to comment on this manuscript. The manuscript titled 'Pollen-based temperature and precipitation changes in the Ohrid Basin (western Balkans) between 160 and 70 ka' covers a sound data set of great value to the palaeoecological community as climate reconstruction for the last interglacial-glacial cycle from this sensitive area are scarce. Overall, I think the work is good and should be published

in CP but there are a number of important details that need to be considered and corrected first. In many instances these are related to terminology, definition of terms and ambiguity or circularity in the phrasing. One important example of this is the use of different nomenclatures, e.g., from alpine region (Riss/Würm glacial), from northern/central Europe (Eemian interglacial), and the special nomenclature from France for interstadials/stadials. Regarding the last interglacial, be careful with the statement that the Eemian was not a stable phase in the Balkan region. The last interglacial at LO clearly shows a classical interglacial with an early warming at the beginning of MIS 5e, a climate optimum and a progressive cooling towards the end of the last interglacial. In general, I suggest to interpret the LO record with regard to further regional climate reconstructions (mentioned in the chapter 'Introduction', e.g., LGdM and Ioannina) and use it as a basis for discussing in more global scale with possible correlations to the France, speleothem records, MIS, etc. These important issues and more are detailed below along with some suggestions for grammatical corrections. Page, Line. Comment

Referee: P1, 17. The presented archive covers the period between 160 to 70ka. This includes not only MIS 6 and MIS 5, but also the early part of MIS 4. The authors mentioned it in the conclusion by themselves. (P14, 8).

Authors: OK

P1, 25-28. According to the anonymous referee #1. The last interglacial at Lake Ohrid shows a classical interglacial cycle with pre-temperate phase (early warming), temperate phase (climate optimum), and a post-temperate phase (progressive cooling), which is also well described by Tzedakis, 2007. Be careful with this general conclusion of an unstable last interglacial in the Balkan region!

OK, WE WILL ADJUST THE TEXT

P2, 3. Insert '...'MIS' – 6 (penultimate glacial) and MIS 5 (last interglacial complex) are. . ..'

OK

P2, 6. '. . .the penultimate glacial (or Riss Glaciation) . . .' in comparison to P2, 12. 'The Eemian. . .' Please pay more attention to a uniform nomenclature. The term 'Riss glaciation' is normally used in the alpine area. In northern and central Europe, the penultimate glacial belongs to the 'Late Saale/Saalian Complex'. If you want to continue with the term 'Eemian' for the last interglacial (MIS 5e), you should use the nomenclature of the northern and central Europe. Another example is the '(Early) Weichselian glacial' instead of 'Early Würm' (P2, 34).

OK

P2, 7. Please pay more attention to uniformity. The LIC lasts from ca. 130-80 ka in the chapter 'Introduction', whereas the LIC covers the period from ca. 128-70 ka at page 3, 25. Please check the dates, there are several more discrepancies.

OK, WE MADE IT CLEARER. THE MIS 6 TO MIS 5 TRANSITION AT LAKE OHRID HAS BEEN THE SUBJECT OF AN ACCURATE ALIGNMENT AND SYNCHRONIZA-TION (ZANCHETTA ET AL., 2016) RESULTING IN AN OFFSET OF 2 KA WITH OTHER RECORDS (E.G. GOVIN ET AL., 2015; RAILSBACK ET AL., 2015).

P3, 2-4. The authors mentioned that Ioannina and LGdM have done climate reconstructions based on pollen data. Unfortunately, these two archives were not used for comparison (e.g. in Figure 5) or were not discussed in detail in the text (chapter 5.3), although these records are much closer to Lake Ohrid than the archives in France. Due to the fact that you mentioned in P3, line 15 that the Balkan Peninsula is a key region between the Mediterranean area and the Northern/Central Europe. It would be nice to see how these few southern European records differ from the northern European ones. What about the direct comparison with Lake Prespa, which covers the last âĹij90 ka. I am not sure if they have done climate reconstructions, but are there any similarities or differences to your record?

THE IOANNINA AND MONTICCHIO CLIMATE RECONSTRUCTIONS HAVE BEEN DONE BY BREWER ET AL. (2008); WE HAVE NOT USED THEM FOR COMPARISON IN FIGURE 5 BECAUSE THIS STUDY ONLY FOCUSES ON THE EEMIAN PART, BUT WE HAVE USED THESE SITES IN FIGURE 6 (EEMIAN PART) FOR COMPARISON WITH OHRID (SOUTH EUROPE CLIMATE CURVE IS BASED ON IOANNINA, MON-TICCHIO AND 2 MARINES CORES); SO, YES WE HAVE USED IOANNINA AND MONTICCHIO SYNTHETIC CLIMATE RECONSTRUCTIONS TO COMPARE OUR DATA WITH A MORE GENERAL SOUTH EUROPEAN CURVE (FIG 6). THERE IS NOT A QUANTITATIVE CLIMATE RECONSTRUCTION FOR LAKE PRESPA, AND THE CHRONOLOGY FOR BOTTOM CORE IS NOT WELL CONSTRAINED.

P3, 14. . . .glacial-interglacial cycle. (?)

OK

P4, 1-2. The first sentence is not necessary.

OK

P4, 11. Avoid the repetition of 'karst aquifers' at the end of line 11.

OK

P4, 12-13. Rephrase: '. . .small streams, rivers, and by direct precipitation.'

OK

P4, 18. Rephrase: '...during winter and south-southeasterly (or southerly to south-easterly) winds during. . ..'

OK

P4, 21. This context is not clear – please rephrase. What are the four zones? Which species dominate which zone?

OK, WE WILL CORRECT IT

P5, 1-3. Please check. This sentence is written in a different language.

OK, WE WILL DELETE IT

P5, 15. What is 'new' in the high-resolution pollen data, presented in this manuscript, when it is already published in Sinopoli et al., 2018? Did you analyse more 'new' samples for this manuscript, which are not shown in the Sinopoli et al. paper? Please clarify!

OK, WE WILL MAKE IT CLEARER.

P5, 34. Which 'six modern analogues'? This subject should be further explained and clarified in the text.

SEE THE ANSWER TO REVIEW 1, WE WILL CHANGE THE TEXT ACCORDINGLY.

P6, 18. It is not clear to me what do you mean with the '. . .first analogue and the last analogue. . .'? More details are needed.

THE FIRST ANALOGUE CORRESPONDS TO THE CLOSEST ANALOGUE, BASED ON THE CHORD DISTANCE CALCULATION; THE LAST ONE IS THE ANALOGUE WITH THE CHORD DISTANCE. WE WILL CORRECT IN THE TEXT AND IN FIGURE A2

P6, 35-36. Please check. This sentence is written in a different language.

OK, WE WILL DELETE IT

P7, 1. '. . .and annual precipitation between 350 and 600 mm/yr),. . .' It depends on what method you looking at. For MAT, I can recognize a fluctuation from 100 to ca. 300 mm/yr in the mean annual precipitation. For WALPS, it fluctuates between 500 to 700 mm/yr. What is the explanation for this huge difference? Please clarify and add some more explanations.

THESE DIFFERENCES ARE DISCUSSED IN THE DISCUSSION PART: DURING

THE FIRST PART OF MIS 6 (CA. 160-143 KA), EVEN IF THE PRECIPITATION CURVES PRODUCED BY BOTH METHODS SHOW THE SAME TREND, RECON-STRUCTED VALUES BY MAT ARE ROUGHLY 300 MM LOWER THAN THOSE RE-SULTING FROM WAPLS. MODERN ANALOGUES METHODS ARE VERY SENSI-TIVE TO MINOR VARIATIONS IN THE POLLEN ASSEMBLAGES, ESPECIALLY DUR-ING GLACIAL PERIODS (BREWER ET AL., 2008). SIMILAR DISCREPANCIES AS-SOCIATED WITH MAT ALSO OCCUR IN THE RECONSTRUCTION OF LA GRANDE PILE AFTER THE EEMIAN THERMAL OPTIMUM BY BREWER ET AL (2008). AM-BIGUOUS OUTCOMES MAY OCCUR PARTICULARLY FOR PAST GLACIAL AND COLD INTERVALS (STADIALS). THE MAJOR PROBLEM APPEARS TO BE THE LACK OF MODERN ANALOGUES OR ONLY LIMITED SIMILARITY WITH PAST GLACIAL VEGETATION (GUIOT ET AL., 1993; PEYRON ET AL., 1998). INDEED, AS REPORTED IN SEVERAL STUDIES (GUIOT, 1987; GUIOT ET AL., 1993; KLOTZ ET AL., 2003), GLACIAL STEPPE VEGETATION DOMINATED BY HIGH PERCENT-AGES OF AMARANTHACEAE (AS AT LAKE OHRID, FIG. 2) HAS NO PRESENT-DAY ANALOGUE IN EUROPE. FOR THIS REASON, WE HAVE USED MODERN SAM-PLES FROM COLD STEPPE, PRINCIPALLY FROM THE TIBETAN PLATEAU AND FROM RUSSIA AS "POTENTIAL" ANALOGUES FOR GLACIAL PERIODS (PEYRON ET AL., 1998, 2005) AND THESE SAMPLES ARE CHARACTERIZED BY LOW PRE-CIPITATIONS VALUES.

P7, 11 and 14. Which 'other methods'? If necessary, add references.

WE ARE NOT SURE WE SHOULD LIST ALL OF THEM, AS WE REFER TO THE ORIGINAL PUBLICATIONS (BREWER ET AL, 2008 AND KUHL ET AL 2010 FOR MULTI-METHOD APPROACHES); WE WILL ADD THE PDF METHOD IN THE TEXT.

P7, 34-36. Which is the third part? Furthermore, an additional verb is missing. Please rephrase this sentence.

MELISEY II AS SHOWN IN TABLE 1, YES "ARE" IS MISSING. WE WILL ADJUST IT

P8, 17 and following. Describe the 'end of MIS 6' within chapter 4.1

YES, HERE IS JUST USED AS A COMPARISON

P9, 8 -16. This section should be mentioned in the chapter 'Materials & methods'.

WE DON'T AGREE: THIS SECTION TRIES TO EXPLAIN THE DIFFERENCES IN THE RESULTS, DUE TO THE DIFFERENT METHODS; SO IT'S BETTER TO KEEP IT IN THE DISCUSSION PART.

P9, 19-20. To be consistent with the text, could you add the discussed pioneer shrubs (e.g. Juniperus) to the selected pollen diagram (Figure 2).

OK

P9, 28 and following. For the better understanding and demonstration, it would be very helpful to show the comparison of your Eemian climate reconstructions with those from the JO2004 record. Please insert the JO2004 climate reconstruction, for example, in Figure 4.

THE CLIMATE RECONSTRUCTION BASED ON THE JO2004 RECORD DONE WITH A MULTI-METHOD APPROACH (PEYRON ET AL., PERS. COMM.) IS NOT YET PUB-LISHED; SO IT'S NOT POSSIBLE TO INCLUDE IT FOR COMPARISON IN FIG .4.

P10, 12. The phrase '. . .export of terrestrial organic matter. . .' sounds a bit odd. Please rephrase.

OK

P10, 18 and 19. Delete '. . .inferred from pollen.' due to the repetition from the previous sentence. At this point, it is obvious that TANN and PANN were calculated from pollen.

OK

P10, 18-29. There seem to be some logical steps missing. I cannot work out how lake level changes can be visible in the pollen record. I also cannot see a decline

in terrestrial vegetation at the end of MIS 6 - in fact, quite the opposite. It shows a continuous increase in mesophilous and coniferous trees! In addition, I assume that significant lake level changes should be reflected in the TIC /TOC values, but again I cannot see any changes in these proxies at the DEEP site. Furthermore, the 'clearly seen' change in the pollen record of Co2012 is not presented in this manuscript! These subjects should be explained more clearly in the text.

THE REFEREE IS RIGHT, LAKE LEVEL CHANGES ARE NOT VISIBLE IN THE POLLEN RECORD! WE USED OTHER EVIDENCES FROM PUBLISHED ARTICLES: "THE DISTINCT HIGH-AMPLITUDE FLUCTUATIONS INFERRED FROM POLLEN DURING THE FINAL PART OF MIS 6, COULD AT LEAST PARTLY BE DUE TO LAKE LEVEL CHANGES AS THE WATER TABLE DURING THIS PERIOD WAS GENER-ALLY ON THE RISE (LINDHORST ET AL., 2010, HOLTVOETH ET AL., 2017; WAG-NER ET AL., 2017)". THIS IS IN VERY GOOD AGREEMENT WITH "A CONTINUOUS INCREASE IN MESOPHILOUS AND CONIFEROUS TREES" AS EVIDENCED BY REFEREE 2. ANYWAY, IT APPEARS THAT WE HAVE NOT BEEN CLEAR ENOUGH, WE WILL TRY TO BE MORE EFFECTIVE.

P10, 31-34. In my opinion, a difference of 500 years is NOT a discrepancy. The authors should soften the language.

WE THOUGHT THAT USING "SLIGHT" COULD HAVE BEEN ENOUGH, AND IT IS NOT FOR REFEREE 2... BUT REFEREE 1 SAYS IS NOT THAT SLIGHT... SO THE OPPOSITE "THIS SLIGHT DISCREPANCY" (NAMELY 500 YEARS) "IS PROBABLY DUE TO DIFFERENCES IN THE CHRONOLOGY ESTABLISHED FOR THE TWO CORES". WE WILL EMPHASIZE THAT THE TWO CHRONOLOGIES HAVE BEEN ASSESSED INDEPENDENTLY.

P10, 39-41. Please avoid the use of so many 'and' in this sentence. Please rephrase.

OK

P10, 42 and following. There is something odd about the line of reasoning here. It is not clear to me what do the authors mean with 'from 120 ka and culminating at 119.4 ka'? In Fig. 4, I cannot identify a 'culmination' in the TIC decrease during this interval. At the DEEP site, the TIC and TOC values already continuously decrease after ca. 126 ka! In addition, how can a progressive drying (P11, 3) take place when precipitation increases at the same time (P11, 1)? By the way, I cannot see an increase in precipitation after 120 ka! Please clarify.

OK, WE WILL CLARIFY THIS PART IN THE TEXT

Chapter 5.3. It would be nice to see a direct comparison of climate parameters between Lake Ohrid, LGdM, Ioannina, and the records in France (e.g., in Figure 5). Unfortunately, the southern European records are only summarized in Figure 6. I think it would be helpful for your argumentation. Be careful with simplification of complex interactions! When I am looking at the comparison between LO, Northern Europe, and Southern Europe (Figure 6), I can recognize several different responses to global climate changes in all records. In my opinion, the authors should make it unequivocally clear the transitional position from Mediterranean climate influenced climate to more temperate northern European climate conditions with, e.g., a distinct temperature decrease after 125 ka, which is not that pronounced at LO (more comparable to the southern European records).

WE AGREE WITH YOU. THE IOANNINA AND MONTICCHIO CLIMATE RECON-STRUCTIONS HAVE BEEN DONE BY BREWER ET AL. (2008); FOR IOANNINA NO SINGLE CURVE FOR CLIMATE RECONSTRUCTION IS AVAILABLE, THE CLIMATE RECONSTRUCTIONS HAVE BEEN USED TO BE INCLUDED AS SUMMARY CURVE FOR SOUTHERN EUROPEAN SITES. MOREOVER, WE CANNOT USE THEM FOR COMPARISON IN THE FIGURE 5 BECAUSE THIS STUDY ONLY FOCUSES ON THE EEMIAN PART. THEREFORE, WE HAVE USED THESE SITES IN THE FIG-URE 6 (EEMIAN PART) FOR COMPARISON WITH OHRID (SOUTH EUROPE CLI-MATE CURVE IS BASED ON IOANNINA, MONTICCHIO AND 2 MARINES CORES);

SO, YES WE HAVE USED IOANNINA AND MONTICCHIO CLIMATE RECONSTRUC-
TIONS TO COMPARE OUR DATA WITH THE MORE GENERAL SOUTH EUROPEAN
CURVES (FIG 6).

P11, 26. The period from 135-105 ka comprises the late MIS 6 to MIS 5c, as you
already mentioned it in the next sentence!

YES, THE REFEREE IS RIGHT. WE USED IN FACT THE VERB "INCLUDES", MEAN-
ING THAT IS NOT ONLY THE WHOLE EEMIAN: "THE PERIOD 135- 105 KA, WHICH
INCLUDES THE WHOLE EEMIAN"

P11, 29. In the direct comparison between LO and Grande Pile & Bouchet, there are
opposite trends in the anomalies at the end of MIS 6. Between ca. 140-133 ka: high
anomalies at LO, low at GP; between ca. 133-128 ka: low anomalies at LO, high at
GP. Please clarify.

OK, DUE TO THE PROBLEMS ALREADY LISTED IN HAVING PRESENT-DAY ANA-
LOGUES FOR AMARANTHACEAE STEPPE, IT'S NOT APPROPRIATE TO GO IN
THESE DETAILS

P12, 11. Delete the repetition of 'Fig.5'.

OK

P12, 18. As I already wrote above, add the 'other pollen records from Lake Ohrid' to
the figures. It would be helpful for the following of your argumentation.

THE REFEREE IS RIGHT, BUT A COMPARISON OF THE DIFFERENT POLLEN
RECORDS (NOT ALL AVAILABLE TO US) FROM LAKE OHRID IS NOT THE TOPIC
OF THIS PAPER AND JUSTIFY A SEPARATE PUBLICATION

P12, 24-35. What is the 'striking feature' of these interstadials, just the occurrence?
Add some more explanations. I think these two short-term interstadial can be corre-
lated with the Dansgaard-Oeschger events DO 19 and 20, which are also visible in

the eastern Mediterranean records, such as Thenaghi Philippon (Müller et al., 2011) and Lake Van (Pickarski et al., 2015), even though the climate was significantly more continental during this time.

YES, THEY CAN BE PROBABLY CORRELATED WITH THE DANSGAARD-OESCHGER EVENTS DO 19 AND 20. WE DO NOT UNDERSTAND "EVEN THOUGH THE CLIMATE WAS SIGNIFICANTLY MORE CONTINENTAL DURING THIS TIME", MAYBE THE REFEREE INTENDS IN THAT REGION.

P13, 17 & 23. The ODP-976 record is not presented in the manuscript!

YES, THE REFEREE IS RIGHT, IT'S A MISTAKE

P13, 16-24. There are some important differences visible between LO and other records (e.g., at ODP-977, Villars cave) especially at the early Eemian, which are not discussed in the text. Be careful with generalization! Please add more details and discussed that differences a bit more.

YES, THERE ARE DIFFERENCES, WE ARE AWARE OF, DIFFICULT TO EXPLAIN. IT ANYWAY GOES BEYOND THE SCOPE OF THIS PAPER TO EXPLAIN ALL DIFFERENCES WITH OTHER RECORDS IN DETAIL.

P13, 25 -34. There are some logical steps missing. Which event centered at ca. 115? Melisey I? C25? I am a bit lost in this section! In addition, C25 event is not visible in the SST record! Please clarify!

OK, WE WILL CLARIFY THIS PART.

P14, 5. A period is missing at the end of the sentence.

OK, WE WILL CLARIFY THIS PART.

P14, 8. Insert '...Last Interglacial Complex (LIC, 128 to 70 ka),...' due to the used abbreviation in the next sentence.

OK. JUST IN CASE SOME READERS ARE JUST READING THE CONCLUSIONS.

P14, 12. '. . .occurring during the late MIS 6, MIS 5 and the early MIS 4.' Table 1 Please use a uniform nomenclature. It would be nice if you could mark the different MIS 5 stages (MIS 5e to a) in the 'Marine Statigraphy' column.

WE WILL DO A ROUGH SCHEME, BUT AS FAR AS WE KNOW THE DIRECT COR-RESPONDENCE BETWEEN TERRESTRIAL PHASES AND MARINE STRATIGRA-PHY WAS NOT YET PRECISELY ESTABLISHED.

Figures Figure 1 Where is the 'Struga meterological station' located? Can you mark it on the map? Please pay more attention to the consistence of facts between the text and the figures. For example, you mentioned in P4, 7 that Lake Ohrid is located at 693 m asl. In your figure 1, it is written 694 m asl. The same discrepancy is evident in the mean annual temperature at Lake Ohrid (P4, line 15).

OK, THANK YOU

Figure 2 Perhaps it is better to use the terms 'Mesophilous taxa/biome' and 'Mediter-ranean taxa/biome' instead of 'trees', because Hedera is not a tree, it is a liane, and Cistus (depending on the species) grows also as shrub. The figure caption is a bit confusing. If you are showing, e.g., only Poaceae within the group of grasslands than delete the additional information that grasslands consist of Poaceae and Cyperaceae. The same goes for 'Steppe'. Please, show in the first column (left) the MIS 6 to 4 and in the second column the nomenclature of the northern and central Europe. That goes also for the other figures.

WE USED THE SAME CATEGORIES AND NAMES USED IN SINOPOLI ET AL. 2018 AND SADORI ET AL. 2016 FOR UNIFORMITY.

Figure 3 MAT method is shown in a blue line, not in black! Regarding GDD5: The legend of the figure is not clear to me. Are these 1000-3000 years over 5◦C per year/season/? Are these 1000-3000◦C. Please, clarify! Delete the repetition of

'Blue shading indicates cold periods (Riss glacial and Early Würm glacial stadials)' in the figure caption.

OK, WE WILL ADJUST IT; GDD5, THE GROWING DEGREES DAYS OVER 5 IS DEFINED AS THE SUM OF POSITIVE TEMPERATURE (PER DAY) OVER A PERIOD (YEAR) ABOVE A CERTAIN THRESHOLD BASE TEMPERATURE (HERE 5°C)

Figure A1 What do you mean with 'The last graph represents the . . ..'? Figure A2? What outline the different red lines? More details are needed.

IT'S A MISTAKE: THE SENTENCE "THE LAST GRAPH REPRESENTS..." CORRESPONDS TO THE CAPTION OF THE FIGURE A2; WE WILL CORRECT IT. IN THE FIGURE CAPTION OF FIG. A2, TEXT WILL BE ADDED FOR CLARITY: THE FIRST ANALOGUE CORRESPONDS TO THE BEST OR CLOSEST ANALOGUE WITH THE LOW CHORD DISTANCE; THE LAST ANALOGUE CORRESPONDS TO THE ANALOGUE WITH THE HIGHER CHORD DISTANCE

References Müller et al., 2011. The role of climate in the spread of modern humans into Europe, QSR 30. 273-279. Pickarski et al., 2015. Abrupt climate and vegetation variability of eastern Anatolia during the last glacial. CP 11. 1491-1505. Rasmussen et al., 2014. A stratigraphic framework for abrupt climatic changes during the Last Glacial period based on three synchronized Greenland ice-core records: refining and extending the INTIMATE event stratigraphy. QSR 106. 14-28. Tzedakis, 2007. Seven ambiguities in the Mediterranean palaeoenvironmental narrative. QSR 26. 2042-2066. I hope my comments help improving the manuscript.

YES, THEY DID. THANK YOU VERY MUCH

---

## Author Comment (AC7) · 7 Nov 2018

After a debate among authors, we decide to reply more in detail to the referees clarifying how we intend to improve the manuscript. We would ask the editor and the referees to consider this version.

The quantitative palaeoclimate reconstructions during the Quaternary is an important to understand the climate changes and its potential forcing mechanisms, thus can helpful for predicting climate changes in future global warming. In this study, a quantitative reconstruction of climate parameters was provided based on the pollen data from the Lake Ohrid in southern Europe, using two complementary approaches including

'Mod- ern Analogues Technique' and 'Weighted Averaging Partial Least-Squares Re-gressio' during the period from 160 to 70 ka. It is useful for better understanding climatic changes during the key periods of MIS 6 and MIS 5 in the South Europe. In current version I would suggest a minor revision before accepting it for publication. C1 Here are a few basic comments that could guide the authors to submit a more de-tailed manuscript. 1.For the reconstruction, Pinus has been excluded in this study due to its overwhelming presence in the DEEP would potentially masks climatically con-trolled environmental signals from other taxa. Because this change should effect on the quantitative reconstruction, so a more detail comparison on reconstruction of cli-mate parameters between Pinus including and its excluding is best presented in the supplementary information.

OF COURSE, OUR FIRST TEST WAS WITH PINUS. AS SUGGESTED BY THE RE-VIEWER, WE WILL ADD, IN THE SUPPLEMENTARY INFORMATION, A FIGURE ON THE CLIMATE RECONSTRUCTION WITH PINUS INCLUDED. WE DON'T DISCUSS IT BECAUSE THE RESULTS ARE CLEARLY LESS GOOD THAN THE FINAL RE-SULTS OBTAINED EXCLUDING PINUS. WE ATTACH A PRELIMINARY VERSION OF THE FIGURE.

2.In Figure A2, the most values of Squared-chord distance between the first and the last analogue for a chosen climate parameter (TANN) calculated by MAT method are more than 0.3. The values may be larger than the no-analog/analog thresh- old that could accurate and precise palaeoclimate reconstruction. Therefore, a systematically analysis between the Squared-chord distance and precision of palaeoclimate recon-struction need to employ.

WE ARE NOT SURE TO UNDERSTAND WHAT THE REVIEWER MEANS: THE DIFFERENCE BETWEEN THE FIRST AND THE LAST ANALOGUE IS CLOSE TO 0.1, NOT TO 0.3. THE THRESHOLD FOR THE ANALOGUES SELECTION IS DE-FINED BY THE METHOD (SEE GUIOT ET AL. FOR DETAILS), AND HERE NO NO-ANALOGUE SITUATION HAS BEEN DETECTED.

3. Results suggest that the Lake Ohrid palaeoclimate reconstruction shows greater similarity with climate patterns inferred from northern European pollen records than with southern European records in figure 6. Because the Lake Ohrid locates in the southern European, thus a more detail explanation and possible mechanism should be mentioned.

WE DID IT, PLEASE LOOK MORE IN DEPTH THE DISCUSSION. FOR EXAMPLE: "THIS SIMILARITY IS PROBABLY DUE TO ITS HIGH ELEVATION, CAUSING EN-HANCED PRECIPITATION IN RELATIVE TO THE REST OF SOUTHERN EUROPE AND MAKING IT SIMILAR TO REGIONS DIRECTLY SUBJECTED TO THE NORTH-ATLANTIC CIRCULATION" AND "THE CONNECTION BETWEEN LAKE OHRID AND THE NORTH ATLANTIC (FIG. 7), IS ALSO HIGHLIGHTED BY THE EVIDENCE OF THE MELISEY I STADIAL, WHICH CORRESPONDS TO NORTH ATLANTIC EVENT C24 (AND TO GS25), THE MONTAIGU EVENT, CORRESPONDING TO C23 (AND GS24), AND THE MELISEY II STADIAL, WHICH CORRESPONDS TO C21 (AND GS22)".

4. Please check all the language in the text, and correct it to the English, e.g. page 5 lines 1-3, and page 6 lines 35-36.

SURE, THANK YOU

———————————————————

[Figure]

**Fig. 1.** Climate parameters obtained including Pinus

---

## Author Response (AR3)

[revised manuscript text omitted]

**REPLY TO EDITOR**

Dear authors,

Thank you for your revised document. I have studied the corrections and see hat lot of the request of reviewers have been

5  addressed in this version. before accepting your manuscript, I request few little amendments:

1 - You present the curve of ODP Site 977 for comparison with your data. Please take care of the typing of the name of this site in the text and in the figures. It is not correctly written everywhere. For example see Fig. 1 (in the draw) and Fig.7 (it is here corrected in the caption but not in the draw).

10  Authors: THANK YOU, WE CORRECTED THEM

in addition in the Figure 7, I think that it is not the right citation for the ODP 977 curve. In Martrat et al., 2014, the curve presented did not cover the whole interval showing here but only from 150kyr to 108 kyr. The paper from the same authors published in 2004 and 2008 are more appropriate here. Please, have a look on the best paper for the citation.

15  THE EDITOR IS RIGHT, WE CHANGE THE CITATION IN THE CAPTION AND IN THE TEXT

2- In table 1, is it possible to add some ages on the left to have some "tie-points"for the different events. Please do that.
WE ADDED THE LAKE OHRID LIMITS

20  3- I agree with the reviewer concerning the groups cited in the Fig. 2. Hedera is not a tree, same for Cistus. Even if it corresponds to groups published in another publication, I suggest to change that. One way may be to replace "trees" by "forest" or "vegetation" or "formation".
WE FOLLOWED THE EDITOR REQUEST

25  4- Please add for GDD5 (..... 5°C/yr) in fig.3
DONE

These changes do not take lot of time to do. So I am waiting after your corrected manuscript to be able to conclude the review process quickly.

30  With my best regards
Nathalie Combourieu-Nebout

THANK YOU FOR IMPROVEMENTS AND CORRECTIONS

**REPLY TO REFEREE 1**

5   After a debate among authors, we decide to reply more in detail to the referees clarify- ing how we intend to improve the manuscript. We would ask the editor and the referees to consider this version.

This is a very interesting paper worthy to be published and here are my suggestions for some improvements following the continuity of the text. Referee: P1, Line 25 (and p.14 lines 13-17). For me, the discussion on the stability of the Eemien rests on a misdefinition. Since Jessen, everybody know that an interglacial cycle includes a pe- riod of warming after the previous

10   glacial, an optimum and finally a progressive cooling

leading to the next glacial. In this way the instability you mention is a truism! You could just mention that "The Eemian in the Balkans was characterized by an abrupt early warming during its anathermic phase followed a central phase . . ..). Most of the authors, when dealing with "instability" or "stability" try to identify some short terms climate "oscillations" disturbing the classical interglacial trend on a warming followed by a cooling (as well discussed latter in your paper).

Authors: THE REFEREE IS RIGHT SAYING THAT THE "INSTABILITY" OF THE EEMIAN IS A WELL ASSESSED SUBJECT. WE CORRECTED IT.

P2, l.30-35. Before Field et al. (1994), at least Beaulieu & Reille (1984,1889) already mentioned a period of transitional

20   warming during the late Eemian.

WE CONSIDERED THIS

P3, l.2 and fig. 6: be careful with the coordinates. Les Echets: 45∘52'36"N, 4∘55'44"E and Lac du Bouchet: 44∘54'31"N ,

25   3∘47'30" Lac du Bouchet is really transitional be- tween North and South according to your classification. As a matter of fact, this limit at 45∘N is of interest as far as the Eemian is concerned, but during the Early Würm the story is more complex with the opposition between the "Odderade style" vegetation and climate successions and the "Grande Pile style" successions. It could be mentioned.

30   THANK YOU, WE CONSIDERED THIS

P3, l.27. : I should prefer "higher resolution", as an interval of 400 is not high resolution in terms of vegetation dynamics. Would you accept a pollen diagram covering the whole Holocene in only 28 spectra?

WE AGREE, IT'S "HIGHER RESOLUTION"

P4, the first sentence is not necessary as the following description is sufficient.

OK

P4, l.11: do not repeat "karst aquifers".

OK

P5, top: my copy is polluted by three lines in pseudo-latin.

WE DELETED THE ODD PHRASE

P5, l.34: could you develop and explain in more details your choice of 6 modern analogues?

TEXT WAS CHANGED

P6, l. 20: I suggest "pollen records " instead of "data" P6, l.35: again one line polluted with latin.

WE CHANGED INTO LOW-RESOLUTION DATA, THE POLLEN RECORD IS THE SAME

P9, l.5 and after: this interesting discussion could be included in the chapter "Materials and methods"?

WE DON'T AGREE: THIS SECTION TRY TO EXPLAIN THE DIFFERENCES IN THE RESULTS DUE TO THE DIFFERENT METHODS USED; SO IT'S REALLY BETTER TO KEEP IT IN THE DISCUSSION PART

P9, l.22: may-be a clumsy statement. If your climate reconstructions are derived from pollen spectra, it would be a great disaster for your results if they were not in agreement

WE DELETED POLLEN

P10, l. 24-29: Very interesting but I do not understand how the discussion is inferred from fig. 4: TIC and TOC do not change (slight increase in TOC) during the interval between 137 and 135 Ka marked by high amplitude changes in PAN and TAN??

WE ARE USING DATA FROM OTHER CORES AND OTHER PAPERS. GENERALLY, LAKE-LEVEL CHANGE RESPONDS TO PRECIPITATION CHANGE RATHER THAN TEMPERATURE AND MAY THEREFORE HAVE AFFECTED THE DISTRIBUTION OF LOWLYING TERRESTRIAL HABITATS ON FLAT TERRACE SURFACES BEFORE THE TEMPERATURE THRESHOLD WAS CROSSED THAT LED TO INCREASING IN LAKE TIC

5    PRECIPITATION. TOC FROM COMBINED AQUATIC AND TERRESTRIAL SOURCES IS A MINOR FRACTION OF THE TOTAL SEDIMENT AND, AS SUCH, IS SUSCEPTIBLE TO DILUTION BY LITHOGENIC MATERIAL. THUS, IF PRECIPITATION INCREASED THE SUPPLY OF BOTH TERRESTRIAL OC AND LITHOGENIC (SILICICLASTIC) MATERIAL WOULD HAVE INCREASED. DEPENDING ON THE MAIN OC SOURCES THIS CAN LEAD TO BOTH AN INCREASE OR DECREASE IN TOC WITH INCREASING EROSION RATES, OR TOC MAY IN FACT REMAIN

10   TEMPORARILY CONSTANT IF ITS MAIN SOURCE IS, E.G., SOIL OC (INCREASE OF OC AND SILICICLASTIC MATERIAL AT SAME RATE). THUS, WE DON'T NECESSARILY EXPECT TO SEE A COVARIATION OF PANN AND TANN WITH TIC AND TOC AT THIS TRANSITIONAL STAGE. WE WILL TRY TO MAKE IT MORE CLEAR

     P10, l 34 : Not that slight??

     SEE COMMENTS OF REFEREE 2, QUITE THE OPPOSITE

     Chapter 5.4. : it would be of interest to take into account the climate reconstructions based on diatoms populations established by Rioual et al. (2007) at Ribains (see also Shemesh et al., 2001)

     THANKS FOR THE SUGGESTION. HOWEVER IT'S NOT POSSIBLE TO ADD IT IN THE FIGURES (E.G. FIG. 5) BECAUSE IN THE RIOUAL ET AL (2007) PAPER, THE TEMPERATURE RECONSTRUCTION BASED ON THE DIATOMS IS NOT DIRECTLY COMPARABLE TO THE OHRID ONES. RIOUAL ET AL (2007) PLOTTED PDCA SAMPLE SCORES POLLEN DIATOMS AS A PROXY OF TEMPERATURE, NOT TEMPER- ATURE VALUES. IN THE

25   PAPER OF RIOUAL ET AL (2001), THE CLIMATE RECONSTRUCTION BASED ON RIBAINS POLLEN DATA CANNOT BE ADDED HERE IN OUR FIGURE AS ITS PLOTTED AS FUNCTION OF DEPTH, NOT OF AGE. BUT, OF COURSE, THESE TWO PAPERS ARE CITED AND USED IT IN OUR INTERPRETATION

     References Beaulieu J.-L. de & Reille M., 1984. A long upper Pleistocene pollen record from Les Echets near Lyon, France,
30   Boreas, 13, p.111-132. Beaulieu J.-L. de & Reille M., 1989. The transition from temperate phases to stadials in the long Upper Pleistocene sequence from les Echets (France). Palaeogeography, ,Palaeocli- matology, Palaeoecology, 72, 147-159. Rioual P.., Andrieu-Ponel V., Beaulieu J.-L. de, Reille M., Svobodova H. & Battarbee R. W., 2007. Diatom responses to limnological and climatic changes at Ribains maar (French Massif Central) during the Eemian and Early Würm. Quaternary Science Reviews, 26 (11-12), 1557-1609. Shemesh A., Rietti-Shati M., Rioual P., Battarbee R., Beaulieu J.-L. de, Reille M. and

Svobodova H., 2001. An Oxygen isotope record of lacustrine opal from a european Maar indicates climatic stability during the last interglacial. Geophysical Research Let- ters, 28 (12), 2305-2308.

**REPLY TO REFEREE 2**

After a debate among authors, we decide to reply more in detail to the referees clarifying how we intend to improve the manuscript. We would ask the editor and the referees to consider this version.

Thank you for the opportunity to comment on this manuscript. The manuscript titled 'Pollen-based temperature and precipitation changes in the Ohrid Basin (western Balkans) between 160 and 70 ka' covers a sound data set of great value to the palaeoecological community as climate reconstruction for the last interglacial-glacial cycle from this sensitive area are scarce. Overall, I think the work is good and should be published in CP but there are a number of important details that need to be considered and cor- rected first. In many instances these are related to terminology, definition of terms and ambiguity or circularity in the phrasing. One important example of this is the use of different nomenclatures, e.g., from alpine region (Riss/Würm glacial), from north- ern/central Europe (Eemian interglacial), and the special nomenclature from France for interstadials/stadials. Regarding the last interglacial, be careful with the statement that the Eemian was not a stable phase in the Balkan region. The last interglacial at LO clearly shows a classical interglacial with an early warming at the beginning of MIS 5e, a climate optimum and a progressive cooling towards the end of the last interglacial. In general, I suggest to interpret the LO record with regard to further regional climate reconstructions (mentioned in the chapter 'Introduction', e.g., LGdM and Ioannina) and use it as a basis for discussing in more global scale with possible correlations to the France, speleothem records, MIS, etc. These important issues and more are detailed below along with some suggestions for grammatical corrections. Page, Line. Comment

Referee: P1, 17. The presented archive covers the period between 160 to 70ka. This includes not only MIS 6 and MIS 5, but also the early part of MIS 4. The authors mentioned it in the conclusion by themselves. (P14, 8).

Authors: OK

P1, 25-28. According to the anonymous referee #1. The last interglacial at Lake Ohrid shows a classical interglacial cycle with pre-temperate phase (early warming), temper- ate phase (climate optimum), and a post-temperate phase (progressive cooling), which is also well described by Tzedakis, 2007. Be careful with this general conclusion of an unstable last interglacial in the Balkan region!

WE ADJUSTED THE TEXT

P2, 3. Insert '...'MIS' – 6 (penultimate glacial) and MIS 5 (last interglacial complex) are. . ..'

OK

5 P2, 6. '. . .the penultimate glacial (or Riss Glaciation) . . .' in comparison to P2, 12. 'The Eemian. . .' Please pay more attention to a uniform nomenclature. The term 'Riss glaciation' is normally used in the alpine area. In northern and central Europe, the penultimate glacial belongs to the 'Late Saale/Saalian Complex'. If you want to continue with the term 'Eemian' for the last interglacial (MIS 5e), you should use the nomenclature of the northern and central Europe. Another example is the '(Early) Weichselian glacial' instead of 'Early Würm' (P2, 34).

OK

P2, 7. Please pay more attention to uniformity. The LIC lasts from ca. 130-80 ka in the chapter 'Introduction', whereas the LIC covers the period from ca. 128-70 ka at page 3, 25. Please check the dates, there are several more discrepancies.

OK, WE MADE IT CLEARER. THE MIS 6 TO MIS 5 TRANSITION AT LAKE OHRID HAS BEEN THE SUBJECT OF AN ACCURATE ALIGNMENT AND SYNCHRONIZATION (ZANCHETTA ET AL., 2016) RESULTING IN AN OFFSET OF 2 KA WITH OTHER RECORDS (E.G. GOVIN ET AL., 2015; RAILSBACK ET AL., 2015).

20 P3, 2-4. The authors mentioned that Ioannina and LGdM have done climate reconstructions based on pollen data. Unfortunately, these two archives were not used for comparison (e.g. in Figure 5) or were not discussed in detail in the text (chapter 5.3), although these records are much closer to Lake Ohrid than the archives in France. Due to the fact that you mentioned in P3, line 15 that the Balkan Peninsula is a key region between the Mediterranean area and the Northern/Central Europe. It would be nice to see how these few southern European records differ from the northern European ones. What about 25 the direct comparison with Lake Prespa, which covers the last ấLij90 ka. I am not sure if they have done climate reconstructions, but are there any similarities or differences to your record?

THE IOANNINA AND MONTICCHIO CLIMATE RECONSTRUCTIONS HAVE BEEN DONE BY BREWER ET AL. (2008); WE HAVE NOT USED THEM FOR COMPARISON IN FIGURE 5 BECAUSE THIS STUDY ONLY FOCUSES 30 ON THE EEMIAN PART, BUT WE HAVE USED THESE SITES IN FIGURE 6 (EEMIAN PART) FOR COMPARISON WITH OHRID (SOUTH EUROPE CLIMATE CURVE IS BASED ON IOANNINA, MONTICCHIO AND 2 MARINES CORES); SO, YES WE HAVE USED IOANNINA AND MONTICCHIO SYNTHETIC CLIMATE RECONSTRUCTIONS TO COMPARE OUR DATA WITH A MORE GENERAL SOUTH EUROPEAN CURVE (FIG 6). THERE IS NOT A QUANTITATIVE CLIMATE RECONSTRUCTION FOR LAKE PRESPA, AND THE CHRONOLOGY FOR BOTTOM 35 CORE IS NOT WELL CONSTRAINED.

P3, 14. . . .glacial-interglacial cycle. (?)

OK

P4, 1-2. The first sentence is not necessary.

OK

P4, 11. Avoid the repetition of 'karst aquifers' at the end of line 11.

OK

P4, 12-13. Rephrase: '. . .small streams, rivers, and by direct precipitation.'

OK

P4, 18. Rephrase: '...during winter and south-southeasterly (or southerly to south- easterly) winds during. . ..'

OK

P4, 21. This context is not clear – please rephrase. What are the four zones? Which species dominate which zone?

WE CORRECTED IT

P5, 1-3. Please check. This sentence is written in a different language.

WE DELETED IT

P5, 15. What is 'new' in the high-resolution pollen data, presented in this manuscript, when it is already published in Sinopoli et al., 2018? Did you analyse more 'new' samples for this manuscript, which are not shown in the Sinopoli et al. paper? Please clarify!

WE MADE IT CLEARER.

P5, 34. Which 'six modern analogues'? This subject should be further explained and clarified in the text.

SEE THE ANSWER TO REVIEW 1, WE CHANGED THE TEXT ACCORDINGLY.

P6, 18. It is not clear to me what do you mean with the '. . .first analogue and the last analogue. . .'? More details are needed.

THE FIRST ANALOGUE CORRESPONDS TO THE CLOSEST ANALOGUE, BASED ON THE CHORD DISTANCE CALCULATION; THE LAST ONE IS THE ANALOGUE WITH THE CHORD DISTANCE. WE WILL CORRECT IN THE TEXT AND IN FIGURE

P6, 35-36. Please check. This sentence is written in a different language.

WE DELETED IT

P7, 1. '. . .and annual precipitation between 350 and 600 mm/yr),. . .' It depends on what method you looking at. For MAT, I can recognize a fluctuation from 100 to ca. 300 mm/yr in the mean annual precipitation. For WALPS, it fluctuates between 500 to 700 mm/yr. What is the explanation for this huge difference? Please clarify and add some more explanations.

THESE DIFFERENCES ARE DISCUSSED IN THE DISCUSSION PART: DURING THE FIRST PART OF MIS 6 (CA. 160-143 KA), EVEN IF THE PRECIPITATION CURVES PRODUCED BY BOTH METHODS SHOW THE SAME TREND, RECONSTRUCTED VALUES BY MAT ARE ROUGHLY 300 MM LOWER THAN THOSE RESULTING FROM WAPLS. MODERN ANALOGUES METHODS ARE VERY SENSITIVE TO MINOR VARIATIONS IN THE POLLEN ASSEMBLAGES, ESPECIALLY DURING GLACIAL PERIODS (BREWER ET AL., 2008). SIMILAR DISCREPANCIES ASSOCIATED WITH MAT ALSO OCCUR IN THE RECONSTRUCTION OF LA GRANDE PILE AFTER THE EEMIAN THERMAL OPTIMUM BY BREWER ET AL (2008). AMBIGUOUS OUTCOMES MAY OCCUR PARTICULARLY FOR PAST GLACIAL AND COLD INTERVALS (STADIALS). THE MAJOR PROBLEM APPEARS TO BE THE LACK OF MODERN ANALOGUES OR ONLY LIMITED SIMILARITY WITH PAST GLACIAL VEGETATION (GUIOT ET AL., 1993; PEYRON ET AL., 1998). INDEED, AS REPORTED IN SEVERAL STUDIES (GUIOT, 1987; GUIOT ET AL., 1993; KLOTZ ET AL., 2003), GLACIAL STEPPE VEGETATION DOMINATED BY HIGH PERCENTAGES OF AMARANTHACEAE (AS AT LAKE OHRID, FIG. 2) HAS NO PRESENT-DAY ANALOGUE IN EUROPE. FOR THIS REASON, WE HAVE USED MODERN SAMPLES FROM COLD STEPPE, PRINCIPALLY FROM THE TIBETAN PLATEAU AND FROM RUSSIA AS "POTENTIAL" ANALOGUES FOR GLACIAL PERIODS (PEYRON ET AL., 1998, 2005) AND THESE SAMPLES ARE CHARACTERIZED BY LOW PRECIPITATIONS VALUES.

P7, 11 and 14. Which 'other methods'? If necessary, add references.

WE ARE NOT SURE WE SHOULD LIST ALL OF THEM, AS WE REFER TO THE ORIGINAL PUBLICATIONS (BREWER ET AL, 2008 AND KUHL ET AL 2010 FOR MULTI-METHOD APPROACHES); WE ADDED THE PDF METHOD IN THE TEXT.

P7, 34-36. Which is the third part? Furthermore, an additional verb is missing. Please rephrase this sentence.

MELISEY II AS SHOWN IN TABLE 1, YES "ARE" IS MISSING. WE ADJUSTED IT

P8, 17 and following. Describe the 'end of MIS 6' within chapter 4.1

YES, HERE IS JUST USED AS A COMPARISON

P9, 8 -16. This section should be mentioned in the chapter 'Materials & methods'.

WE DON'T AGREE: THIS SECTION TRIES TO EXPLAIN THE DIFFERENCES IN THE RESULTS, DUE TO THE DIFFERENT METHODS; SO IT'S BETTER TO KEEP IT IN THE DISCUSSION PART.

P9, 19-20. To be consistent with the text, could you add the discussed pioneer shrubs (e.g. Juniperus) to the selected pollen diagram (Figure 2).

OK

P9, 28 and following. For the better understanding and demonstration, it would be very helpful to show the comparison of your Eemian climate reconstructions with those from the JO2004 record. Please insert the JO2004 climate reconstruction, for example, in Figure 4.

THE CLIMATE RECONSTRUCTION BASED ON THE JO2004 RECORD DONE WITH A MULTI-METHOD APPROACH (PEYRON ET AL., PERS. COMM.) IS NOT YET PUBLISHED; SO IT'S NOT POSSIBLE TO INCLUDE IT FOR COMPARISON IN FIG .4.

P10, 12. The phrase '. . .export of terrestrial organic matter. . .' sounds a bit odd. Please rephrase.

OK

P10, 18 and 19. Delete '. . .inferred from pollen.' due to the repetition from the previous sentence. At this point, it is obvious that TANN and PANN were calculated from pollen.

OK

P10, 18-29. There seem to be some logical steps missing. I cannot work out how lake level changes can be visible in the pollen record. I also cannot see a decline in terrestrial vegetation at the end of MIS 6 - in fact, quite the opposite. It shows a continuous increase in mesophilous and coniferous trees! In addition, I assume that significant lake level changes should be reflected in the TIC /TOC values, but again I cannot see any changes in these proxies at the DEEP site. Furthermore, the 'clearly seen' change in the pollen record of Co2012 is not presented in this manuscript! These subjects should be explained more clearly in the text.

THE REFEREE IS RIGHT, LAKE LEVEL CHANGES ARE NOT VISIBLE IN THE POLLEN RECORD! WE USED OTHER EVIDENCES FROM PUBLISHED ARTICLES: "THE DISTINCT HIGH-AMPLITUDE FLUCTUATIONS INFERRED FROM POLLEN DURING THE FINAL PART OF MIS 6, COULD AT LEAST PARTLY BE DUE TO LAKE LEVEL CHANGES AS THE WATER TABLE DURING THIS PERIOD WAS GENER- ALLY ON THE RISE (LINDHORST ET AL., 2010, HOLTVOETH ET AL., 2017; WAG- NER ET AL., 2017)". THIS IS IN VERY GOOD AGREEMENT WITH "A CONTINUOUS INCREASE IN MESOPHILOUS AND CONIFEROUS TREES" AS EVIDENCED BY REFEREE 2. ANYWAY, IT APPEARS THAT WE HAVE NOT BEEN CLEAR ENOUGH, WE WILL TRY TO BE MORE EFFECTIVE.

P10, 31-34. In my opinion, a difference of 500 years is NOT a discrepancy. The authors should soften the language.

WE THOUGHT THAT USING "SLIGHT" COULD HAVE BEEN ENOUGH, AND IT IS NOT FOR REFEREE 2. . . BUT REFEREE 1 SAYS IS NOT THAT SLIGHT. . . SO THE OPPOSITE "THIS SLIGHT DISCREPANCY" (NAMELY 500 YEARS) "IS PROBABLY DUE TO DIFFERENCES IN THE CHRONOLOGY ESTABLISHED FOR THE TWO CORES". WE WILL EMPHASIZE THAT THE TWO CHRONOLOGIES HAVE BEEN ASSESSED INDEPENDENTLY.

P10, 39-41. Please avoid the use of so many 'and' in this sentence. Please rephrase.

OK

P10, 42 and following. There is something odd about the line of reasoning here. It is not clear to me what do the authors mean with 'from 120 ka and culminating at 119.4 ka'? In Fig. 4, I cannot identify a 'culmination' in the TIC decrease during this interval. At the DEEP site, the TIC and TOC values already continuously decrease after ca. 126 ka! In addition, how can a

progressive drying (P11, 3) take place when precipitation increases at the same time (P11, 1)? By the way, I cannot see an increase in precipitation after 120 ka! Please clarify.

OK, WE CLARIFIED THIS PART IN THE TEXT

Chapter 5.3. It would be nice to see a direct comparison of climate parameters between Lake Ohrid, LGdM, Ioannina, and the records in France (e.g., in Figure 5). Unfortunately, the southern European records are only summarized in Figure 6. I think it would be helpful for your argumentation. Be careful with simplification of complex interactions! When I am looking at the comparison between LO, Northern Europe, and Southern Europe (Figure 6), I can recognize several different responses to global climate changes in all records. In my opinion, the authors should make it unequivocally clear the transitional position from Mediterranean climate influenced climate to more temperate northern European climate conditions with, e.g., a distinct temperature decrease after 125 ka, which is not that pronounced at LO (more comparable to the southern European records).

WE AGREE WITH YOU. THE IOANNINA AND MONTICCHIO CLIMATE RECONSTRUCTIONS HAVE BEEN DONE BY BREWER ET AL. (2008); FOR IOANNINA NO SINGLE CURVE FOR CLIMATE RECONSTRUCTION IS AVAILABLE, THE CLIMATE RECONSTRUCTIONS HAVE BEEN USED TO BE INCLUDED AS SUMMARY CURVE FOR SOUTHERN EUROPEAN SITES. MOREOVER, WE CANNOT USE THEM FOR COMPARISON IN THE FIGURE 5 BECAUSE THIS STUDY ONLY FOCUSES ON THE EEMIAN PART. THEREFORE, WE HAVE USED THESE SITES IN THE FIG- URE 6 (EEMIAN PART) FOR COMPARISON WITH OHRID (SOUTH EUROPE CLIMATE CURVE IS BASED ON IOANNINA, MONTICCHIO AND 2 MARINES CORES); SO, YES WE HAVE USED IOANNINA AND MONTICCHIO CLIMATE RECONSTRUCTIONS TO COMPARE OUR DATA WITH THE MORE GENERAL SOUTH EUROPEAN CURVES (FIG 6).

P11, 26. The period from 135-105 ka comprises the late MIS 6 to MIS 5c, as you already mentioned it in the next sentence!

YES, THE REFEREE IS RIGHT. WE USED IN FACT THE VERB "INCLUDES", MEANING THAT IS NOT ONLY THE WHOLE EEMIAN: "THE PERIOD 135-105 KA, WHICH INCLUDES THE WHOLE EEMIAN"

P11, 29. In the direct comparison between LO and Grande Pile & Bouchet, there are opposite trends in the anomalies at the end of MIS 6. Between ca. 140-133 ka: high anomalies at LO, low at GP; between ca. 133-128 ka: low anomalies at LO, high at GP. Please clarify.

OK, DUE TO THE PROBLEMS ALREADY LISTED IN HAVING PRESENT-DAY ANALOGUES FOR AMARANTHACEAE STEPPE, IT'S NOT APPROPRIATE TO GO IN THESE DETAILS

P12, 11. Delete the repetition of 'Fig.5'.

OK

P12, 18. As I already wrote above, add the 'other pollen records from Lake Ohrid' to the figures. It would be helpful for the following of your argumentation.

THE REFEREE IS RIGHT, BUT A COMPARISON OF THE DIFFERENT POLLEN RECORDS (NOT ALL AVAILABLE TO US) FROM LAKE OHRID IS NOT THE TOPIC OF THIS PAPER AND JUSTIFY A SEPARATE PUBLICATION

P12, 24-35. What is the 'striking feature' of these interstadials, just the occurrence? Add some more explanations. I think these two short-term interstadial can be correlated with the Dansgaard-Oeschger events DO 19 and 20, which are also visible in the eastern Mediterranean records, such as Thenaghi Philippon (Müller et al., 2011) and Lake Van (Pickarski et al., 2015), even though the climate was significantly more continental during this time.

YES, THEY CAN BE PROBABLY CORRELATED WITH THE DANSGAARD- OESCHGER EVENTS DO 19 AND 20.

P13, 17 & 23. The ODP-976 record is not presented in the manuscript!

YES, THE REFEREE IS RIGHT, IT'S A MISTAKE, WE CORRECTED IT

P13, 16-24. There are some important differences visible between LO and other records (e.g., at ODP-977, Villars cave) especially at the early Eemian, which are not discussed in the text. Be careful with generalization! Please add more details and discussed that differences a bit more.

YES, THERE ARE DIFFERENCES, WE ARE AWARE OF, DIFFICULT TO EXPLAIN. IT ANYWAY GOES BEYOND THE SCOPE OF THIS PAPER TO EXPLAIN ALL DIFFERENCES WITH OTHER RECORDS IN DETAIL.

P13, 25 -34. There are some logical steps missing. Which event centered at ca. 115? Melisey I? C25? I am a bit lost in this section! In addition, C25 event is not visible in the SST record! Please clarify!

OK, WE WILL CLARIFY THIS PART.

P14, 5. A period is missing at the end of the sentence.

OK, WE CLARIFIED THIS PART.

P14, 8. Insert '...Last Interglacial Complex (LIC, 128 to 70 ka),...' due to the used abbreviation in the next sentence.

5    OK. JUST IN CASE SOME READERS ARE JUST READING THE CONCLUSIONS.

P14, 12. '. . .occurring during the late MIS 6, MIS 5 and the early MIS 4.' A Table 1 Please use a uniform nomenclature. It would be nice if you could mark the different MIS 5 stages (MIS 5e to a) in the 'Marine Statigraphy' column.

10   WE DID IT

Figures Figure 1 Where is the 'Struga meterological station' located? Can you mark it on the map? Please pay more attention to the consistence of facts between the text and the figures. For example, you mentioned in P4, 7 that Lake Ohrid is located at 693 m asl. In your figure 1, it is written 694 m asl. The same discrepancy is evident in the mean annual temperature at Lake
15   Ohrid (P4, line 15).

OK, THANK YOU

Figure 2 Perhaps it is better to use the terms 'Mesophilous taxa/biome' and 'Mediter- ranean taxa/biome' instead of 'trees',
20   because Hedera is not a tree, it is a liane, and Cistus (depending on the species) grows also as shrub. The figure caption is a bit confusing. If you are showing, e.g., only Poaceae within the group of grasslands than delete the additional information that grasslands consist of Poaceae and Cyperaceae. The same goes for 'Steppe'. Please, show in the first column (left) the MIS 6 to 4 and in the second column the nomenclature of the northern and central Europe. That goes also for the other figures.

25   WE USED THE SAME CATEGORIES AND NAMES USED IN SINOPOLI ET AL. 2018 AND SADORI ET AL. 2016 FOR UNIFORMITY.

Figure 3 MAT method is shown in a blue line, not in black! Regarding GDD5: The legend of the figure is not clear to me. Are these 1000-3000 years over 5âU ̊e ̧C per year/season/? Are these 1000-3000âU ̊e ̧C. Please, clarify! Delete the repetition of

**REPLY TO REFEREE 3**

After a debate among authors, we decide to reply more in detail to the referees clarifying how we intend to improve the manuscript. We would ask the editor and the referees to consider this version.

5 The quantitative palaeoclimate reconstructions during the Quaternary is an important to understand the climate changes and its potential forcing mechanisms, thus can helpful for predicting climate changes in future global warming. In this study, a quantitative reconstruction of climate parameters was provided based on the pollen data from the Lake Ohrid in southern Europe, using two complementary approaches including

10 'Modern Analogues Technique' and 'Weighted Averaging Partial Least-Squares Regressio' during the period from 160 to 70 ka. It is useful for better understanding climatic changes during the key periods of MIS 6 and MIS 5 in the South Europe. In current version I would suggest a minor revision before accepting it for publication. Here are a few basic comments that could guide the authors to submit a more detailed manuscript. 1.For the reconstruction, Pinus has been excluded in this study due to its overwhelming presence in the DEEP would potentially masks climatically controlled environmental signals from other

15 taxa. Because this change should effect on the quantitative reconstruction, so a more detail comparison on reconstruction of climate parameters between Pinus including and its excluding is best presented in the supplementary information.

OF COURSE, OUR FIRST TEST WAS WITH PINUS. AS SUGGESTED BY THE REVIEWER, WE ADDED, IN THE SUPPLEMENTARY INFORMATION, A FIGURE (FUIGURE A) ON THE CLIMATE RECONSTRUCTION WITH

20 PINUS INCLUDED. WE DON'T DISCUSS IT BECAUSE THE RESULTS ARE CLEARLY LESS GOOD THAN THE FINAL RESULTS OBTAINED EXCLUDING PINUS.

2.In Figure A2, the most values of Squared-chord distance between the first and the last analogue for a chosen climate parameter (TANN) calculated by MAT method are more than 0.3. The values may be larger than the no-analog/analog thresh-

25 old that could accurate and precise palaeoclimate reconstruction. Therefore, a systematically analysis between the Squared-chord distance and precision of palaeoclimate reconstruction need to employ.

WE ARE NOT SURE TO UNDERSTAND WHAT THE REVIEWER MEANS: THE DIFFERENCE BETWEEN THE FIRST AND THE LAST ANALOGUE IS CLOSE TO 0.1, NOT TO 0.3. THE THRESHOLD FOR THE ANALOGUES

30 SELECTION IS DE- FINED BY THE METHOD (SEE GUIOT ET AL. FOR DETAILS), AND HERE NO NO-ANALOGUE SITUATION HAS BEEN DETECTED.

3. Results suggest that the Lake Ohrid palaeoclimate reconstruction shows greater similarity with climate patterns inferred from northern European pollen records than with southern European records in figure 6. Because the Lake Ohrid locates in the southern European, thus a more detail explanation and possible mechanism should be mentioned.

WE DID IT, PLEASE LOOK MORE IN DEPTH THE DISCUSSION. FOR EXAMPLE: "THIS SIMILARITY IS PROBABLY DUE TO ITS HIGH ELEVATION, CAUSING ENHANCED PRECIPITATION IN RELATIVE TO THE REST OF SOUTHERN EUROPE AND MAKING IT SIMILAR TO REGIONS DIRECTLY SUBJECTED TO THE NORTH-ATLANTIC CIRCULATION" AND "THE CONNECTION BETWEEN LAKE OHRID AND THE NORTH ATLANTIC (FIG. 7), IS ALSO HIGHLIGHTED BY THE EVIDENCE OF THE MELISEY I STADIAL, WHICH CORRESPONDS TO NORTH ATLANTIC EVENT C24 (AND TO GS25), THE MONTAIGU EVENT, CORRESPONDING TO C23 (AND GS24), AND THE MELISEY II STADIAL, WHICH CORRESPONDS TO C21 (AND GS22)".

4. Please check all the language in the text, and correct it to the English, e.g. page 5 lines 1-3, and page 6 lines 35-36.

SURE, THANK YOU

**INTERACTIVE COMMENT OF ATTILA DEMÉNY**

WE THANK ATTILA DEMÉNY FOR HIS COMMENT. WE INCLUDE THE SUGGESTED PROXY IN THE DISCUSSION SECTION.